# AGENTOCCAM: A SIMPLE YET STRONG BASELINE FOR LLM-BASED WEB AGENTS

**Ke Yang**[†,*] **Yao Liu**[◇]**, Sapana Chaudhary**[◇]**, Rasool Fakoor**[◇]**, Pratik Chaudhari**[◇]**, George Karypis**[◇]**, Huzefa Rangwala**[◇]

University of Illinois Urbana-Champaign[†], Amazon[◇]

`key4@illinois.edu`, `{yaoliuai,chausapa,fakoor,rhuzefa}@amazon.com`

## ABSTRACT

Autonomy via agents based on large language models (LLMs) that can carry out personalized yet standardized tasks presents a significant opportunity to drive human efficiency. There is an emerging need and interest in automating web tasks (e.g., booking a hotel for a given date within a budget). Being a practical use case itself, the web agent also serves as an important proof-of-concept example for various agent grounding scenarios, with its success promising advancements in many future applications. Meanwhile, much prior research focuses on handcrafting their web agent strategies (e.g., agent's prompting templates, reflective workflow, role-play and multi-agent systems, search or sampling methods, etc.) and the corresponding in-context examples. However, these custom strategies often struggle with generalizability across all potential real-world applications. On the other hand, there has been limited study on the misalignment between a web agent's observation and action representation, and the data on which the agent's underlying LLM has been pre-trained. This discrepancy is especially notable when LLMs are primarily trained for language completion rather than tasks involving embodied navigation actions and symbolic web elements. In our study, we enhance an LLM-based web agent by simply refining its observation and action space, aligning these more closely with the LLM's capabilities. This approach enables our base agent to significantly outperform previous methods on a wide variety of web tasks. Specifically, on WebArena, a benchmark featuring general-purpose web interaction tasks, our agent AGENTOCCAM surpasses the previous state-of-the-art and concurrent work by 9.8 (+29.4%) and 5.9 (+15.8%) absolute points respectively, and boosts the success rate by 26.6 points (+161%) over similar plain web agents with its observation and action space alignment. Furthermore, on WebVoyager benchmark comprising tasks defined on real-world websites, AGENTOCCAM exceeds the former best agent by 2.4 points (+4.6%) on tasks with deterministic answers. We achieve this without using in-context examples, new agent roles, online feedback or search strategies. AGENTOCCAM's simple design highlights LLMs' impressive zero-shot performance on web tasks, and underlines the critical role of carefully tuning observation and action spaces for LLM-based agents.[1]

## 1 INTRODUCTION

AI agents leveraging large language models (LLMs) show great potential in automating repetitive and programmatic tasks and thereby alleviating human workloads (Gao et al., 2024; Xi et al., 2023; Yang et al., 2024). LLMs showcase remarkable capabilities in perception, reasoning and planning primarily due to their pre-training and post-learning. However, their effectiveness is significantly constrained when task-specific observation and action representations diverge from the parametric knowledge encoded during their training/learning time. For instance, in web-based tasks, these agents perform notably below human levels (Zhou et al., 2023b; Koh et al., 2024a).

To improve web task performance by LLM-based agents, recent work focuses on designing better agent policies with either handcrafted prompting templates (Sodhi et al., 2024) or hard-coded auto-

---

*Work performed while interning at Amazon.

[1]Our code and data are available at https://github.com/amazon-science/AgentOccam.

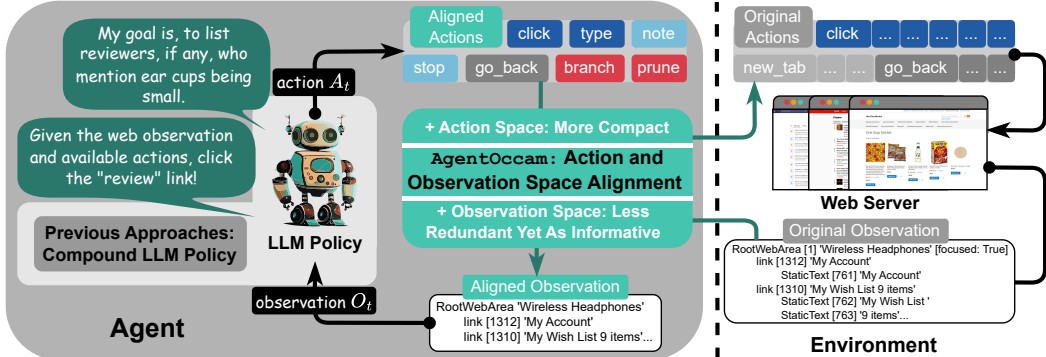

Figure 1: Overview of AGENTOCCAM. Unlike prior research that works intensively on designing compound LLM policies, we enhance the web agent simply by aligning the web interaction action and observation space with the functioning LLM's acquired knowledge and skills during its training.

prompting strategies (Fu et al., 2024; Wang et al., 2024). While those pre-defined strategies can be effective for certain tasks, they struggle to generalize to diverse websites and varying skill requirements. Another emerging trend is to adopt sampling or search algorithms for a dynamic exploration of web navigation actions, which reduces dependence on pre-defined strategies but increases the cost of LLM inferences (Koh et al., 2024b; Zhang et al., 2024; Pan et al., 2024).

In this work, we aim to enhance an LLM-based web agent's proficiency by optimizing the text-based task understanding and reasoning of existing LLMs, rather than refining the agent strategies. Automating web tasks is challenging, as the agent needs to *i)* accurately extract information from web pages with varying formats and encoded scripts, and *ii)* issue appropriate embodied actions, selecting from those defined merely on web (e.g., scrolling, clicking, or hovering over buttons). These web observation and action spaces are less common in both, the pre- and post-training data of LLMs, preventing the LLMs from fully realizing their potential in accomplishing general-purpose web tasks. Therefore, we study how to properly tune the observation and actions for LLM-based web agents, to align them with the functioning LLMs capacities learned during training.

As shown in Figure 1, our method comprises of three components: *i)* We reduce non-essential actions to minimize the agent's embodiment and trivial interaction needs; *ii)* We refine the observation by eliminating redundant and irrelevant web elements, and restructuring web content blocks for more succinct yet as informative representations; *iii)* We introduce two planning actions (`branch` and `prune`), which enables the agent to self-organize navigation workflow with a planning tree, and use the same structure to filter the previous traces for history replay. It's noteworthy that those planning commands are in the same position as navigation prompts for the agent. We implement these components by generic rules that applies to all types of markup-language-formatted web pages, without leveraging task-related information on the test benchmark.

By combining the three techniques mentioned above, our proposed agent AGENTOCCAM performs substantially better on web tasks across websites in the WebArena environments (Zhou et al., 2023b). AGENTOCCAM outperforms the previous state-of-the-art approach by 9.8 absolute points (+29.4%) and surpasses concurrent work by 5.9 absolute points (+15.8%). Notably, unlike most prior work, we do not use any in-context examples, additional online search or sampling, nor specialized prompting templates or agent roles to play well. In contrast, AGENTOCCAM delivers such strong performance with an unexpectedly simple approach: letting the LLM issue actions within the processed and augmented observation and action spaces. Compared with a similar plain web agent without these proposed observation and action space changes, AGENTOCCAM increases the success rate by 26.6 absolute points (+161%). Moreover, we prove AGENTOCCAM's web environment generalizability in the real-world web environment. In the WebVoyager definite-answer subset (He et al., 2024), which consists of real-world web tasks with deterministic answers, AGENTOCCAM exceeds the previous best agent on this benchmark by 2.4 points (+4.6%).

In summary, the primary contribution of this work are as follows. First, we develop a new state-of-the-art agent, AGENTOCCAM, for web tasks. On the WebArena benchmark consisting of 812 tasks across five diverse websites (e.g., shopping, searching on a forum), AGENTOCCAM outperforms previous and concurrent work significantly. Second, we shed light on the strong zero-shot perfor-

Table 1: Comparison of essential components for different LLM-based web agents.

| Essential Components | Task-specific Strategies | In-context Examples | Additional Module | Offline Data[2] | Online Search |
|---|---|---|---|---|---|
| **AutoGuide (Fu et al., 2024)** | NO | YES | YES | YES | NO |
| **SteP (Sodhi et al., 2024)** | YES | YES | YES | NO | NO |
| **AutoRefine (Pan et al., 2024)** | NO | YES | YES | YES | YES |
| **LM-Tree Search (Koh et al., 2024b)** | NO | YES | YES | YES | YES |
| **AWM (Wang et al., 2024)** | NO | YES | YES | YES[3] | NO |
| **WebPilot (Zhang et al., 2024)** | NO | YES | YES | NO | YES |
| **Agent-E (Abuelsaad et al., 2024)** | NO | YES | YES | NO | NO |
| AGENTOCCAM | NO | NO | NO | NO | NO |

mance of LLMs on web tasks with our simple agentic workflow, in sharp contrast to many more complex compound agent policies. Last, our work on aligning the observation and action spaces is orthogonal to agentic strategies and can be combined with future advances in that aspect.

## 2 RELATED WORK

**LLM-based Web Agent** Advances in large language and multi-modal foundation models have significantly boosted the development of autonomous agents to solve web tasks. Techniques translating LLMs to powerful decision-making agents (Yao et al., 2022b; Shinn et al., 2024) have advanced web agents, inspiring many techniques that design inference time agent strategies. Many prior approaches improve the agent system by designing auxiliary modules with specialized LLMs or roles, aiming to break down complex tasks (Sun et al., 2024; Prasad et al., 2024; Abuelsaad et al., 2024). Other work leverages LLMs to extract common patterns from examples or past experience (Zheng et al., 2023; Fu et al., 2024; Wang et al., 2024). However, this line of work often relies on pre-defined control hierarchy, prompt templates or examples to act accurately in the test environments. For example, SteP (Sodhi et al., 2024) utilizes a stack-based approach for dynamic multi-level control in the web tasks but relies on task-specific atomic policies with environment-related information hard-coded in prompt template. Another line of work focuses on improving web agents' performance by leveraging more online examples from the environments. Many of them (Zhou et al., 2023a; Zhang et al., 2024; Putta et al., 2024) adapt Monte Carlo Tree Search (MCTS) methods, expanding intermediate states (tree nodes) in one task repeatedly by multiple trials over that task. Among them, WebPilot (Zhang et al., 2024) also adds a global optimization layer for high-level planning. Koh et al. (2024b) use a trained value function to guide search and to back-trace on the task execution tree. Auto Eval and Refine (Pan et al., 2024) trains a separate evaluator, and improves the task execution using reflective thinking (Shinn et al., 2024) on past trials in the same task. However, sampling or resetting multiple times in the same task, not only increases the inference cost significantly, but also limits its applicability when failed task is not revocable. As a comparison, we highlight the simplicity of our method and its difference with related agent approaches in Table 1.

**Fine-tuned or Trained Models for Web Tasks** Fine-tuning language or multimodal models for web tasks is another effective approach to enhance decision-making capabilities on the web tasks (Yin et al., 2024; Hong et al., 2024; Lai et al., 2024; Putta et al., 2024). While fine-tuning promises more adaptivity and broader optimization space, the size of task-specific fine-tuned models are typically not comparable with the most powerful closed-source models. There is also some early research that trains models to follow natural language command on the web before LLMs emerged, using semantic parsing (Artzi & Zettlemoyer, 2013), reinforcement learning (Branavan et al., 2009) and imitation learning (Liu et al., 2018; Humphreys et al., 2022). However, those fine-tuned agents, limited by the base model's capacities or training data volume, often fail to match those constructed with LLMs regarding performance or/and generalizability, and is beyond the scope of this work.

**Simulated Web Agent Environments** Web agent development has been supported by increasingly complex web simulators for training and evaluation. These range from basic platforms like MiniWoB (Shi et al., 2017) and its extension MiniWoB++ (Liu et al., 2018), to more sophisticated

---

[2]The offline data refers to a labeled dataset to instill human knowledge into models.

[3]AWM supports two scenarios: in offline scenarios it directly leverage an offline dataset, and in online scenarios it relies on a domain-specific evaluator from Pan et al. (2024) which requires offline data to train.

environments such as WebShop (Yao et al., 2022a), WebArena (Zhou et al., 2023b), and Visual-WebArena (Koh et al., 2024a). These simulators progressively incorporate real-world complexities, from simple form-filling to tasks across multiple full-featured websites. In this work, we focus only on the text modality, and use WebArena to evaluate our method's task success and generalizability as it contains different types of websites and task-intents in a single suite. To further assess AGEN-TOCCAM's generalizabilty, we extend experiments to the real-world web environments, evluated with the tasks and golden answers proposed in WebVoyager (He et al., 2024).

## 3  PROBLEM FORMULATION

We formalize the web interaction process by a Partially Observable Markov Decision Process (POMDP, Littman (2009); Spaan (2012)): $\langle \mathcal{O}, \mathcal{S}, \mathcal{A}, P, R, p_0, \gamma \rangle$. In POMDPs, an observation $o \in \mathcal{O}$ consists of information that the agent receives from the web environment, e.g. HTMLs, as well as any instructions and prompts. In this work, we only consider the text modality. A state $s \in \mathcal{S}$ denotes the whole underlying (unobserved) state of the agent and the environment such that the state transition is Markovian. An action $a \in \mathcal{A}$ is either a command recognized by the web environment, or any other unrecognized token sequence that will lead to a stay in the current state. $P$ denotes a deterministic state transition function that records the change in the webpage state given the current state and agent action. $R$ is the reward function that decides the success or failure of the agent's sequence of actions. $p_0$ denotes the initial state distribution which is uniform over tasks tested. The discounting factor $\gamma$ is typically set to 1 when the reward is only assigned at the end of an agent-web interaction episode.

To solve POMDP, a common goal is to find a decision policy $\pi(a_t|h_t)$ maximizing the expected cumulative reward, where $h_t$ denotes the observation history $\{o_0, o_1, ..., o_t\}$. In LLM-based web agent design, that is translated to designing a policy $\pi(a_t|h_t)$ with the help of one or more base LLM policies $\pi_{\text{LLM}}$ and a set of algorithmic modules. In this work, we work on a special class of policies that can be expressed as: $\pi(g(a_t)|h_t) = \pi_{\text{LLM}}(a_t|f(h_t))$, where $f$ and $g$ are rule-based functions that process the observation (including action instructions) and actions for the LLM policy. We name it the observation and action space alignment problem. Notice that under such a problem setting, all of our changes apply only to the observations and the actions. We emphasize not all agent strategies in previous approaches can be represented in this way. For example, search-based algorithms require a control program on the top to select actions and trigger back-tracing; methods with evaluators, reflective thinking or memory modules also necessitate a managing center to alternate between the main LLM and these helper segments or other role-playing LLMs. In contrast, we aim to answer the following question in our work: **Can we build a strong web agent with the base LLM policy $\pi_{\text{LLM}}$ by optimizing only the observation and action mapping $f$ and $g$?**

## 4  METHOD

Rather than introducing any new modules or hierarchical structures on top of the base LLM, our method focuses on a simple web agent workflow that inputs the web observations to a general-purpose LLM-API and uses the LLM outputs as actions directly. In this section, we describe the process of aligning web tasks, which necessitates embodiment knowledge, with the predominantly static and text-centric nature of LLM training. Section 4.1 discusses our strategies (summarized in Figure 2) for refining the action space to be more compact and reducing the need for the agent's embodiment capabilities. Section 4.2 outlines our methods (summarized in Figure 4) for condensing web content descriptions to be both brief and informative, and identifying key web elements and relevant steps for retention to organize the agent's memory in a pertinent manner.

### 4.1  ACTION SPACE ALIGNMENT

A web agent's action space defines the valid commands it can use to interact with the web environment. The WebArena simulator supports translating three categories of actions into mouse and keyboard operations: basic actions (e.g., `click`, `type`), tab operations (e.g., `tab_focus` for managing active tabs), and page operations (e.g., `go_back` for navigation). These actions are detailed in Appendix A, along with a comparison of our changes to the action space.

Based on our observation of common failure modes in web agents, there exist two key challenges to be addressed by editing the action space: *i)* removing irrelevant actions that LLMs struggle to understand and frequently misuse, and *ii)* improving the memorization and planning ability when

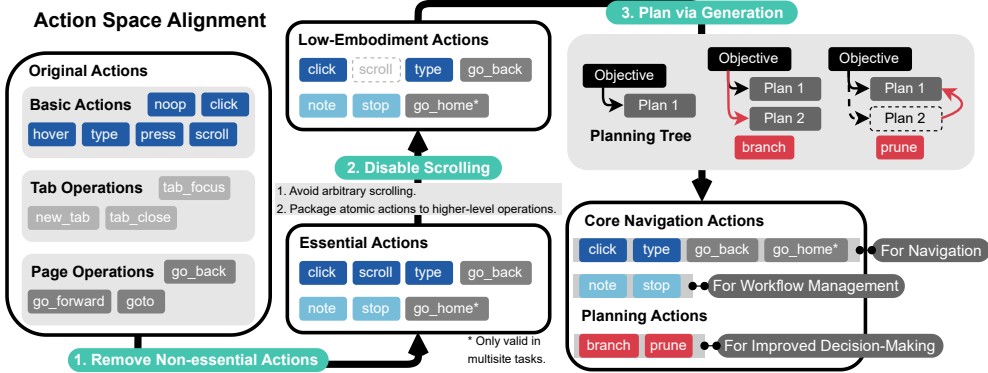

Figure 2: In aligning the action space with LLM pre-training, we only retain high-utility actions and lessen the demand for advanced embodiment skills (steps 1 and 2). Additionally, we incorporate planning steps, allowing the agent to *autonomously* manage task breakdown and execution (step 3).

the task execution requires navigating multiple potential paths to succeed. We propose that the first can be corrected simply by removing and combining actions. The second issue was often addressed in the previous work using handcrafted rules or strategies, making these approaches hard to generalize. In this work, we tackle the second problem by introducing actions that allow the LLM to autonomously generate plans and manage the task workflow. These proposed solutions are explained in detail below and illustrated in Figure 2. The list of all actions in original and reduced action space is shown in Table 3, together with the frequency they are taken in different agents.

**Simplifying the Action Space.** First, we eliminate actions that can be replicated using similar actions or replace multiple actions with one action with the same expressiveness (illustrated in Figure 2 step 1). Specifically, we remove the noop action, signifying "no operation", as it is shown to be a distraction to the agent in most cases. Similarly, tab operations, which manage the focusing, opening, or closing of tabs are removed because they are only needed in limited cases of multi-site tasks requiring two tabs. Furthermore, we limit page navigation actions like go_forward and goto, as their utility is greatly constrained by the agent's poor memory of the relationship between a page's URL and its content. By eliminating these less effective actions, our goal is to minimize distractions and boost the agent's concentration on more meaningful operations. In addition, we introduce the note action, allowing the agent to record key observations for subsequent conclusions, and the stop action, enabling the agent to autonomously conclude the trajectory with answers. We also add a go_home command for multi-site tasks, enabling the agent to navigate directly to the homepage where all available sites are listed.

Second, we eliminate actions that heavily require embodiment knowledge and simplify low-level actions into more abstract operations as shown in Figure 2 step 2. In particular, we reduce commands that LLM-based agents struggle with unless provided with detailed context-specific guidance, like hover or press (the latter is for pressing key combinations, often shortcuts). To properly use these actions requires LLMs to have embodied thinking of the current scenario, especially regarding the mouse position or keyboard operations, which it has not acquired during their training. Additionally, we remove the scroll action, opting instead to load the full page content as the web state. This change is in response to our observation that agents tend to engage in aimless and repetitive scrolling when an essential link is not visible at the top of the page, wasting steps without making progress. Furthermore, we streamline the agent's interaction with drop-down menus; instead of selecting the menu and then an option, a single click command with the ID of the desired option now suffices.

**Planning via Generation.** Web tasks often require a solution that requires navigating multiple paths, e.g. extracting information from one page and submitting it to another page, like the task of creating a refund request on the Contact Us page for a broken product (task template 154), which requires parsing the order ID and refund amount from the order pages. We introduce two actions, branch and prune, to generate plans in a tree structure and save them for future observations. As Figure 2 step 3 shows, the LLM-generated plans start with a root node being the objective of the task. The branch action will generate new subplans under the current node, decomposing high-level objectives into smaller, more manageable subgoals. Conversely, the prune action allows the agent

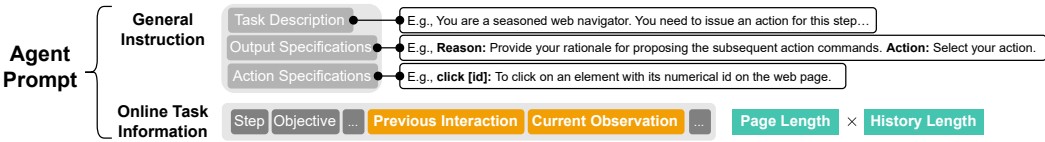

Figure 3: The components of our web navigation agent's prompt. It includes a general instruction outlining the task, the desired output and available actions, as well as online task information providing the current goal, the agent's past interactions, and the latest observations. Notably, the sections on previous interactions and current observation use the most tokens, and can be attributed to two main factors: the length of the pages and the extent of history span.

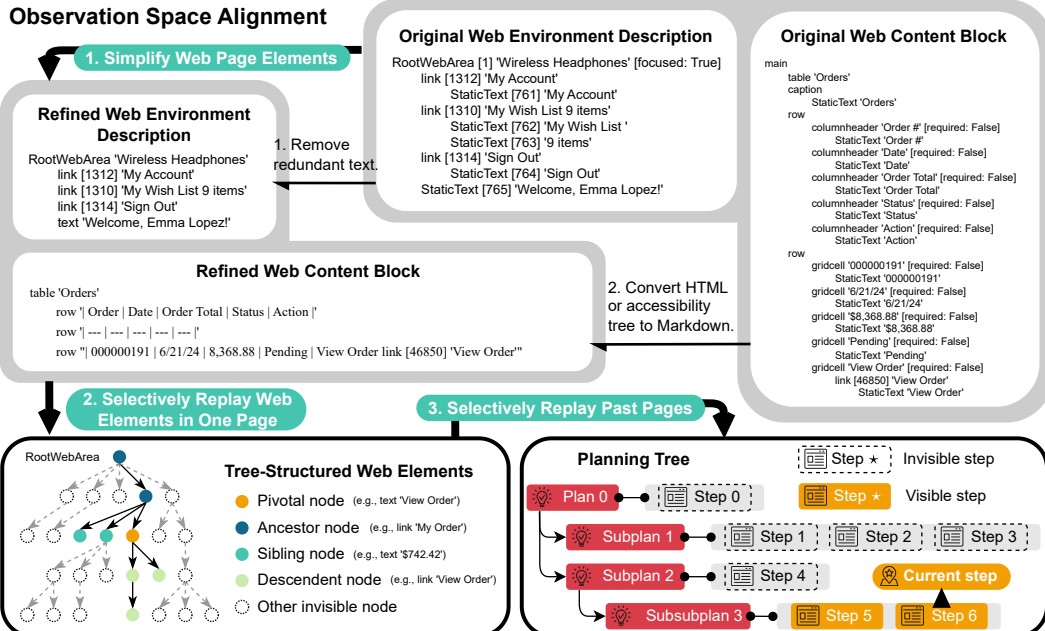

Figure 4: To align the web task's observation space with the format that the base model perceives most effectively, we condense a single-page length by removing unnecessary texts that repetitively describe the web page's functionality and layout (step 1), and by identifying page elements relevant to the task for the agent to remember (step 2). Additionally, we optimize the agent workflow memory through a planning tree, viewing each new plan as a separate goal and excluding past steps' information dedicated to previous plans to enhance memory conciseness (step 3).

to give up the current sub-plan, often after repetitive failed attempts, and seek alternatives. Together with the `branch` and the `prune` actions, the LLM can edit the planning tree autonomously. Note that these two `planning` actions are of no difference from the native `navigation` actions in the web environment (e.g. click, type) and the LLM is free to choose when to take these actions to update the plan. The generated plan provides a context for future action generation and enhances the consistency of actions in one trajectory. This approach leverages the intrinsic planning ability of LLM itself. We argue that this will not compromise the agent's generalization capability as this design relies minimally on task-specific prior knowledge.

## 4.2 Observation Space Alignment

The observation space of web agents consists of task objectives, instructions, previous interaction, and the current web text descriptions or screenshots (see Figure 3 and Appendix G for our agent). Among them, previous interactions and current web content consume the most number of tokens, which scales with the length of a single page and the length of history. This often results in a long context window, which not only increases the LLM inference cost but also poses challenges for LLM to extract related information accurately. Therefore, our main goal in refining the observation is to address these two aspects. The alignment of observations is outlined in Figure 4.

**Simplifying Web Page Observations.** The content on web pages is represented in HTML or accessibility tree format in most text-only web agents. These formats are designed towards front-end loading and rendering, containing numerous formatting tokens making them lengthy and repetitive, as illustrated in Figure 4 Step 1. Our goal is to optimize the representation to make it more readable to LLMs on one single page. Specifically, we merge function-descriptive web elements (e.g., `StaticText [761] 'My Account'`) with interactive elements that share the same label (e.g., `link [1312] 'My Account'`). We then convert table and list blocks to Markdown, eliminating repetitive structural tokens (e.g., `columnheader`, `gridcell`). Consequently, we achieve a more concise representation while keeping the same information.

**Replaying Observation History Selectively.** Taking observation history as input is important for decision-making agents to act consistently for tasks requiring long horizons, given that the observation state only contains partial information about the environment's state. For web tasks, it is also important to include both observation and action history as some key information may not be displayed on the current page. However, the observation history will also significantly scale up the context length and increase reasoning difficulty as well as inference cost. We address this issue by only selecting the most important and related information on the previous web pages, according to two rules based on the "pivotal" nodes (defined later) and the planning tree. We provide detailed examples of how these two techniques are implemented in Appendix B.

First, we observe that only a small amount of content on a web page is pertinent to a specific task among several steps and is worth replaying in future steps. For example, in tasks requiring the agent to find all reviews within three months, it is unnecessary to keep other reviews or some unrelated links like `Contact Us` on the page. Thus we employ a simple rule to identify this small amount of content by leveraging the tree structure of web data (e.g. accessibility tree). To do this, we first instruct the agent to pinpoint the crucial web elements denoted as "pivotal" nodes, at the same time when the agent generates an action. The agent is then programmed to include only the pivotal nodes' ancestor nodes (indicating their global hierarchy and position), sibling nodes (providing immediate context), and descendant nodes (offering detailed characteristics) in the future observations as illustrated in Figure 4 Step 2. This effectively narrows down the volume of data and level of noise passed to the future context of LLM inference.

Second, we observe that not all previous steps' observation needs to be noted during the inference of future steps. Thus we can leverage the planning tree generated by the agent itself to keep the agent's focus sharp. Specifically, when the agent initiates a `branch` action to develop a new plan, we treat this new plan as a separate goal. Steps taken for earlier plans and their observations will be dismissed in the current plan's observation window, as depicted in Figure 4 step 3. This allows the agent to focus only on information dedicated to the current plan for a sub-task.

## 5 EXPERIMENTAL RESULTS AND ANALYSIS

Here, we detail experiments on WebArena (Zhou et al., 2023b), a web simulator benchmark. Further experiments with WebVoyager (He et al., 2024), a web benchmark based on real-world websites, are included in Appendix C. We show AGENTOCCAM's base model generalizability in Appendix D.

**Environment.** We utilize WebArena, an interactive web simulator, as our benchmark. WebArena consists of fully functional websites from four common domains: e-commerce platforms (OneStopShop), social forums for idea and opinion exchange (Reddit), collaborative software development (GitLab), and content management for creation and management of online data (online store management). The platform additionally includes utility tools: a map, a calculator, a scratchpad, and Wikipedia to enable human-like task-solving. The benchmark consists of 812 tasks generated from 241 templates. A template here is a parametric form of a task intent, allowing for multiple instantiations with different keywords. Each task is accompanied by an evaluator that programmatically checks the correctness of the final information with respect to the desired ground truth information[4]. We use `GPT-4-turbo-2024-04-09` (Achiam et al., 2023) to build our AGENTOCCAM.

**Baselines.** We compare AGENTOCCAM with the following prior and concurrent work: 1) WebArena agent: the Chain-of-Thought (CoT) prompted agent included in the WebArena benchmark

---

[4]We identified and corrected errors in the original evaluators, with details discussed in Appendix E. Our approach outperforms the baseline methods with both the original and the corrected evaluators.

Table 2: Comparison of the success rate (SR) of AGENTOCCAM with baseline agents on WebArena.

| Agent | Model | SR (%) (#812) | Shopping (#187) | Shopping Admin (#182) | GitLab (#180) | Map (#109) | Reddit (#106) | Multisite (#48) |
|---|---|---|---|---|---|---|---|---|
| WebArena-replication | GPT-4-Turbo | 16.5 | 16.6 | 15.9 | 10.0 | 22.9 | 21.7 | 16.7 |
| SteP-replication | GPT-4-Turbo | 33.3 | 33.2 | 32.4 | 26.7 | 35.8 | 52.8 | 12.5 |
| AWM | GPT-4 | 35.5 | - | - | - | - | - | - |
| WebPilot | GPT-4o | 37.2 | - | - | - | - | - | - |
| AGENTOCCAM | GPT-4-Turbo | **43.1** | **40.6** | **45.6** | **37.8** | **46.8** | **61.3** | **14.6** |

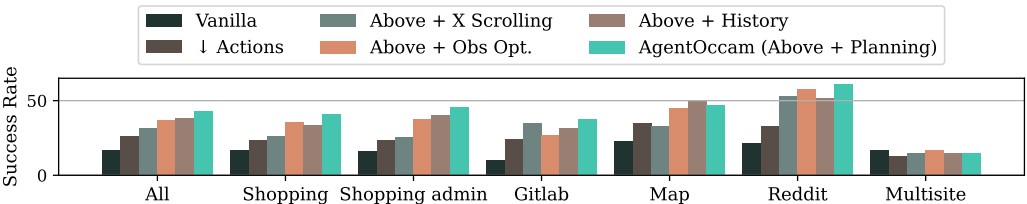

Figure 5: Ablation study of AGENTOCCAM's action and observation space alignment, with details in Table 17. We incrementally add refinement components and assess marginal performance gains.

(Zhou et al., 2023b). 2) SteP (Sodhi et al., 2024): a stack-based approach on top of 14 human-written atomic strategies tailored to solving WebArena. 3) WebPilot (Zhang et al., 2024): a multi-agent, multi-level MCTS based agent that reports state-of-the-art overall performance on WebArena. 4) Agent Workflow Memory (AWM) (Wang et al., 2024): a method automatically summarizing workflow from past experience. SteP has made its code and interaction trajectories public. Hence, we are able to fully replicate the agents from WebArena and SteP with `GPT-4-turbo` in the identical web environments as our methods, for a fair comparison.[5] WebPilot and AWM, being concurrent works with this paper, have not yet provided source code or resulting trajectories, limiting our analysis of these works to just reporting the aggregated performance numbers included in their technical reports. Our analysis focuses on SteP as it was the most performant method prior to this work.

**Question 1: How well does AGENTOCCAM perform?** As seen from the results in Table 2, our agent AGENTOCCAM, which optimizes the action and observation space, now sets a new SOTA on the WebArena benchmark. It increases the overall success rate from 37.2% to 43.1%, a **15.8% relative improvement over best results among previous and concurrent work**. We observe that AGENTOCCAM not only accomplishes tasks in the template that is previously unsolvable, like updating personal information on OneStopShop (task template 165), but it also raises the success rate for templates with mixed results previously, such as setting a homepage URL on a GitLab profile (task template 331). This is further illustrated in Figure 6 in the appendix.

**Question 2: How much does each observation and action space change contribute to AGENTOCCAM?** We evaluate the contribution of each component in AGENTOCCAM described in Section 4 to its overall success by incrementally integrating them into the vanilla agent (WebArena-Replication) and assessing the marginal performance gain shown in Figure 5. The details of each incremental experiment are as follows:

*i)* **Removal of non-essential actions (↓ `Actions`):** Narrowing the action space can reduce the level of distraction for LLM policies and significantly improves performance across all tested websites as shown in Figure 5. By removing rarely used actions like `tab_focus`,`go_forward`, `hover` and `press`, the agent spends fewer steps wandering around and explores more efficiently using actions such as `click` and `type`. Table 3 shows it reduces hundreds of `hover` and `goto` actions while significantly increasing the number of `click` and `type`.

*ii)* **Disabling scrolling (`Above + X Scrolling`):** We observe that LLM policies tend to use `scroll` up and down often when they do not know what to do (since these actions are revertible). Consequently, it significantly delays the task execution and causes looping in certain tasks. As a

---

[5]In our experiments, we note that all agents occasionally fail due to errors from the WebArena simulator, such as posting rate limits in Reddit or login expiration. In such cases, we restart the experiments.

Table 3: Action statistics for the ablation study of AGENTOCCAM's components. Each number in the table represents the frequency of an action across all the tasks within the experiment setting. Actions `noop`, `go_forward`, `tab_focus` and `tab_close` are not included since they are not used even once in vanilla agent and removed in our method.

| Exp. | click | hover | type | press | scroll | new_tab | go_back | goto | note | stop | go_home | branch | prune |
|---|---|---|---|---|---|---|---|---|---|---|---|---|---|
| Vanilla | 2328 | 126 | 1024 | 7 | 132 | 20 | 71 | 511 | - | -[6] | - | - | - |
| ↓ Actions | 7119 | - | 2531 | - | 370 | - | 52 | - | 194 | 512 | 36 | - | - |
| Above + X Scrolling | 7033 | - | 2390 | - | - | - | 100 | - | 219 | 536 | 42 | - | - |
| Above + Obs Opt. | 6890 | - | 2040 | - | - | - | 56 | - | 201 | 571 | 23 | - | - |
| Above + History | 4625 | - | 1286 | - | - | - | 94 | - | 112 | 801 | 54 | - | - |
| AGENTOCCAM | 4720 | - | 1159 | - | - | - | 339 | - | 197 | 769 | 42 | 34 | 47 |

Table 4: Average observation tokens per step across WebArena sites. We use the GPT2 tokenizer from HUGGINGFACE (Radford et al., 2019).

| Exp. | All | Shopping | Shopping Admin | GitLab | Map | Reddit | Multisite |
|---|---|---|---|---|---|---|---|
| Vanilla | 2210.2 | 2272.1 | 2460.2 | 2199.1 | 1883.2 | 2132.4 | 1751.0 |
| ↓ Actions | 1652.0 | 1644.7 | 2133.1 | 1981.3 | 912.0 | 1081.2 | 1296.8 |
| Above + X Scrolling | 3376.2 | 3148.0 | 5403.7 | 3364.9 | 1378.1 | 2603.6 | 1975.5 |
| Above + Obs Opt. | 2891.1 | 1722.5 | 4791.7 | 2560.8 | 1476.4 | 3332.3 | 1619.4 |
| Above + History | 3051.3 | 1802.6 | 5140.2 | 3153.3 | 862.1 | 3156.1 | 2030.3 |
| AGENTOCCAM | 2930.9 | 1634.2 | 4920.7 | 3126.8 | 1056.0 | 3697.8 | 1282.5 |

result, disabling the scrolling action and passing the entire page to the agent proves advantageous, especially for GitLab and Reddit tasks. However, this strategy increases the number of observation tokens, which will be addressed by subsequent refinements.

*iii)* **Simplifying web page elements (`Above + Obs Opt.`)**: We remove redundant text and web format as show in Figure 4 Step 1. This results in fewer tokens in the context window, as outlined in Table 4. It helps the agent focus on web elements crucial to task success across all websites and boosts the performance on all task types, except on GitLab, where this sometimes leads the agent to overlook simpler solutions (task id 394).

*iv)* **Selective replay of web elements in one page (`Above + History`)**: In this experiment, we follow step 2 shown in Figure 4 to add a subset of elements from previous web pages as history. We observe that it allows the agent to avoid repetitive actions in tasks, significantly decreasing the steps needed for task completion as demonstrated in Table 5. However, this addition slightly hurts performance in tasks with dense single-page content or those requiring navigation across multiple pages, as shopping and Reddit tasks success rate drops by 3.2 and 6.0 points, respectively.

*v)* **Planning via generation and selective replay of past pages (AGENTOCCAM; `Above + Planning`)**: We introduce actions `branch` and `prune` to allow the agent to autonomously generate plans and exclude historical steps not in the current sub-plan from the prompt context. This results in performance gains in tasks across nearly all websites, alongside a reduction in the required observation tokens. The actions `branch` and `prune` are both primarily used in correcting a failed strategy and trying an alternative path. For example, in the task of identifying the nearest national park to Boston (task id 265), the agent employs a `branch` action to adopt an alternative search strategy after a failed search attempt. In a GitLab task (task id 563), after multiple failed attempts using the `Create project` button, the agent opts for a `prune` action to explore other methods.

**Question 3: Could the power of AGENTOCCAM be combined with other agentic strategies?**
A natural question to ask next is if we can combine these changes with other common agent strategies or prior work, since the changes in observation and action space are orthogonal and complementary to them. We showcase two example studies to answer this question: one with the SteP method (Sodhi et al., 2024) and another action selection method with LLM-as-a-judge.

The judge method is motivated by our observation of the high variation in the agent's behavior. In some key steps, the agent has a certain probability of generating the correct action but often fails to do so, making it hard for the agent to recover from later pages. For instance, when tasked with identifying the most suitable subreddit for posting (task template 6100), the AGENTOCCAM agent

---

[6]We remove `stop` in the statistics for the vanilla WebArena agent as this action is excluded in their officially defined action space. However, their agent is allowed by code to generate `stop` to end the trajectory.

Table 5: Average number of steps per task across all WebArena sites.

| Exp. | All | Shopping | Shopping Admin | GitLab | Map | Reddit | Multisite |
|---|---|---|---|---|---|---|---|
| Vanilla | 6.2 | 6.2 | 6.6 | 5.9 | 5.7 | 7.4 | 4.4 |
| ↓ Actions | 13.3 | 10.6 | 14.3 | 14.8 | 11.9 | 15.2 | 13.7 |
| Above + X Scrolling | 12.7 | 9.0 | 14.0 | 14.8 | 12.7 | 13.0 | 14.0 |
| Above + Obs Opt. | 12.0 | 8.5 | 13.2 | 15.4 | 10.2 | 12.1 | 13.2 |
| Above + History | 8.6 | 5.6 | 9.6 | 10.3 | 8.3 | 7.6 | 12.9 |
| AGENTOCCAM | 9.0 | 6.7 | 9.2 | 10.8 | 8.5 | 8.6 | 13.4 |

Table 6: Success rate (SR) of AGENTOCCAM combined with agent strategies on WebArena.

| Agent | Model | SR (%) (#812) | Shopping (#187) | Shopping Admin (#182) | GitLab (#180) | Map (#109) | Reddit (#106) | Multisite (#48) |
|---|---|---|---|---|---|---|---|---|
| AGENTOCCAM | GPT-4-Turbo | 43.1 | 40.6 | 45.6 | 37.8 | 46.8 | 61.3 | 14.6 |
| AGENTOCCAM + SteP | GPT-4-Turbo | 41.1 | **46.5** | 36.3 | 36.7 | 47.7 | 50.9 | **18.8** |
| AGENTOCCAM + Judge | GPT-4-Turbo | **45.7** | 43.3 | **46.2** | **38.9** | **52.3** | **67.0** | 16.7 |

tends to hastily choose less relevant subreddits and gets stuck there. To address this, we direct the AGENTOCCAM to generate all possible suitable actions instead of one action at each step. These action candidates are then evaluated by another LLM (`GPT-4-turbo` as well) prompted to play the role of a judge and select the best action. The prompts for the judge are included in Appendix G.

Table 6 shows that a AGENTOCCAM + SteP agent, enhanced with task strategies, outperforms the standalone SteP method but doesn't match AGENTOCCAM's base performance. Additionally, combining AGENTOCCAM with a judge role through an action prediction and selection pipeline rectifies some of the base agent's behavioral misconduct.

By analyzing the trajectories of each method, we observe that task-specific strategies like those introduced in SteP can help when the strategy fits the task requirements. For example, in the task of `"Draft an email to the shop owner via their contact us function for a coupon as {reason}"` (task template 163), the AGENTOCCAM + SteP and SteP agents excel by prompting the agent explicitly not to click the submit button after drafting, where AGENTOCCAM fails to follow. However, for tasks outside the designed strategies, these hints can mislead the agent, leading to a 2-point drop in the overall success rate of AGENTOCCAM + SteP compared to AGENTOCCAM only. An example is task 639, where the agent, guided by SteP's instruction `"Under forums, you will see only a subset of subreddits. To get the full list of subreddits, you need to navigate to the Alphabetical option."`, repetitively navigates away from the appropriate subreddit, and generates reasons for its action selection that `"Clicking on the 'Alphabetical' link will help us access a more comprehensive Reddit list."`, demonstrating how hard-coded strategies can distract the agent and hurt generalizability.

The AGENTOCCAM + Judge agent, combining the AGENTOCCAM's generated action list with the second opinion from an LLM judge increases its overall success rate by 2.6%, by completing tasks where it may well fail due to intermediate decision flaws. For example, in choosing the right subreddit for a post (task template 6100), the base AGENTOCCAM might hastily pick from an initial list, whereas the AGENTOCCAM + Judge agent conducts a thorough search using post keywords or explores the entire forum list before drafting the post. This approach minimizes errors due to rushed decisions, increasing the likelihood of successfully completing task series.

## 6 CONCLUSION

In this paper, we proposed a simple but effective LLM-based web agent AGENTOCCAM that refines its action and observation spaces to be more comprehensible for LLMs primarily trained on static text. Unlike other methods, AGENTOCCAM stands out for its remarkably simple policy workflow, requiring no extra modules, additional LLM calls, or in-context examples. This simplicity does not compromise its performance; AGENTOCCAM surpasses previous and contemporary approaches on WebArena by 9.8 (SteP) and 5.9 (WebPilot) absolute points, respectively. Our results emphasize the importance of maintaining a simple agent architecture for better generalizability, echoing Occam's razor principle. In summary, AGENTOCCAM aims to lay a solid foundation and offer valuable insights for future web agent research and development.

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

## A   COMPARISON OF THE VANILLA AND THE ALIGNED ACTION SPACE

Table 7: The action space in WebArena.

| Category | Action Type | Description |
|---|---|---|
| **Basic Actions** | noop | Do nothing |
| | click(elem) | Click at an element |
| | hover(elem) | Hover on an element |
| | type(elem, text) | Type to an element |
| | press(key_comb) | Press a key combination |
| | scroll(dir) | Scroll up and down |
| **Tab Operations** | tab_focus(index) | Focus on the i-th tab |
| | new_tab | Open a new tab |
| | tab_close | Close current tab |
| **Page Operations** | go_back | Visit the last URL |
| | go_forward | Undo go_back |
| | goto(URL) | Go to URL |

Table 8: The aligned action space of AGENTOCCAM.

| Category | Action Type | Description |
|---|---|---|
| **Basic Actions** | ~~noop~~ | ~~Do nothing~~ |
| | click [id] | Click at an element |
| | ~~hover~~ | ~~Hover on an element~~ |
| | type [id] [content] | Type to an element |
| | ~~press~~ | ~~Press a key combination~~ |
| | ~~scroll~~ | ~~Scroll up and down~~ |
| ~~Tab Operations~~ | ~~tab_focus~~ | ~~Focus on the i-th tab~~ |
| | ~~new_tab~~ | ~~Open a new tab~~ |
| | ~~tab_close~~ | ~~Close current tab~~ |
| **Page Operations** | go_back | Visit the last URL |
| | ~~go_forward~~ | ~~Undo go_back~~ |
| | ~~goto~~ | ~~Go to URL~~ |
| | go_home[7] | Go to home page |
| **Workflow Management** | note [content] | Take notes |
| | stop [answer] | Stop with an answer |
| **Planning Actions** | branch [id] [intent] | Generate a new plan |
| | prune [id] [reason] | Restore to a previous plan |

We list the action space of WebArena and our aligned action space in Table 7 and 8, respectively. In detail, we remove non-essential and embodiment-understanding-required actions like noop and scroll, and add more actions for internal workflow management or autonomous planning control.

## B   PIVOTAL NODES AND THE PLANNING TREE CLARIFICATIONS

To prevent confusion, we separate the following explanations for the independent techniques of "selectively replaying web elements on a page" and "selectively replaying entire pages."

---

[7]Valid only in multisite tasks.

### B.1 SELECTIVELY REPLAYING WEB ELEMENTS ON A PAGE WITH PIVOTAL NODES (FIGURE 4 STEP 2)

#### B.1.1 INTUITION AND BACKGROUND

We could view selectively replaying web elements on a page as a focused memory mechanism, where only the information that's relevant to the task would be recorded and replayed as the agent history traces. As all the web pages could be framed into a (DOM) tree structure, any web elements are nodes in this representation. We define pivotal nodes as the web elements, be it interactable or not, that are potentially useful for completing the task. Those pivotal nodes might present information (e.g., customer reviews in a review summary task) or the agent could interact with them to navigate to crucial states (e.g., the search box and the search button in a search task). We obtain these nodes at each step by prompting the agent to generate the page's highlights based on which they issue the action, or any web elements they will attend to if they fail at future steps and restore to the current state. After identifying the pivotal nodes with the agent's efforts, our code supports automatically parsing the web page's DOM tree and retaining nodes that are associated with the pivotal nodes (i.e., pivotal nodes' ancestor, sibling, and descendant nodes, as shown in Figure 4), to get a more succinct but less noisy version of the observation. This version would be used for constructing the agent's focused working history.

### B.2 PROMPT, PIVOTAL NODE SECTION

```
Generate the response in the following format:
...
OBSERVATION HIGHLIGHT:
List the numerical ids of elements on the current webpage based on which you would issue your
    action. Also include elements on the current webpage you would attend to if you fail in
    the future and have to restore to this step. Don't include elements from the previous
    pages. Select elements at a higher hierarchical level if most their children nodes are
    considered crucial. Sort by relevance and potential values from high to low, and separate
    the ids with commas. E.g., '1321, 52, 756, 838'.
```

### B.3 EXAMPLE TASK OBJECTIVE

What is the email address of the Dean of the School of Engineering at Stanford University?

### B.4 AGENTOCCAM'S WORKFLOW REGARDING THE PIVOTAL NODES

0. The task started at the Google.com, with the observation to be:

```
RootWebArea 'Google'
        link [29] 'About'
        link [30] 'Store'
        link [277] 'Gmail'
        link [275] 'Search for Images'
        button [282] 'Google apps'
        link [152] 'Sign in'
        IframePresentational [153]
        search [6]
                combobox [12] 'Search' [required: False]
                button [294] 'Search by voice'
                button [295] 'Search by image'
                button [272] 'Google Search'
                button [273] "I'm Feeling Lucky"
        contentinfo
                link [83] 'Advertising'
                link [84] 'Business'
                link [85] 'How Search works'
                link [89] 'Our third decade of climate action: join us'
                link [93] 'Privacy'
                link [94] 'Terms'
                button [100] 'Settings'
                        generic [102] 'Settings'
```

1. The agent typed the keyword into the search box and identified element 12 (combobox [12] 'Search' [required: False]) and 272 (button [272] 'Google Search') as the pivotal nodes. The web transit to a Google search page with the searched entries listed. Now, the agent would be prompted

with task history and web observation for issuing the next action, where the task history was constructed based on the pivotal nodes selected by the agent previously, and any pivotal-node-associated nodes. We take the automatically-generated interaction history clip from the prompt at this step:

```
<step_0_interaction>
OBSERVATION:
RootWebArea 'Google'
        search
                combobox 'Search' [required: False]
                button 'Search by voice'
                button 'Search by image'
                button 'Google Search'
                button "I'm Feeling Lucky"
REASON FOR ACTION:
To progress towards finding the email address of the Dean of the School of Engineering at
     Stanford University, the first logical step is to search for relevant information. Using
      the search input box to enter a query such as "Dean of the School of Engineering Stanford
      University email" and then submitting it using the 'Google Search' button is a direct
      approach to gather this information.
ACTION:
type [12] [Dean of the School of Engineering Stanford University email] [1]
</step_0_interaction>
```

2. We can observe that the "pivotal node" technique helps keep a focused memory attention by only retaining the context of crucial information and omitting irrelevant noise. Based on the interaction history summary and current observation, the agent executed the following steps, with the history section constantly updated with the information from the ongoing steps till trajectory ended.

## B.5 Selectively Replaying Entire Pages with the Planning Tree (Figure 4 Step 3)

We will use a planning tree generated by our agent during the development stage as an example for explaining how the planning tree takes shape and helps in selectively replaying past pages.

### B.5.1 Intuition and Background

If we make an analogy of the single-thread web task completion to code execution, the agent's issuing sub plans mimic nested function calling, the active plan is like the uppermost function in the execution stack, and pruning the planning tree plays as popping failed functions off the stack. There constantly exists a planning tree and the planning tree operation instructions in the agent's prompt. We enable the agent to organize the planning tree with 'branch' and 'prune' commands, where the 'branch' action creates new subplans, and the 'prune' action restores the task progress to a previous state. It's noteworthy that those planning commands are in the same position as navigation prompts for the agent.

### B.5.2 Prompt, Planning Tree Section

```
If you think you should refine the plan, use the following actions:
branch [parent_plan_id] [new_subplan_intent]: To create a new subplan based on PREVIOUS PLANS.
     Ensure the new subplan is connected to the appropriate parent plan by using its ID. E.g
     ., 'branch [12] [Navigate to the "Issue" page to check all the issues.]'
prune [resume_plan_id] [reason]: To return to a previous plan state when the current plan is
     deemed impractical. Enter the ID of the plan state you want to resume. E.g., 'prune [5] [
     The current page lacks items "black speaker," prompting a return to the initial page to
     restart the item search.]'
Otherwise, use the following actions:
{navigation_specifications}
```

## B.6 Example Task Objective

(WebArena 174) Open my latest updated issue that has keyword "feature" in its title to check if it is closed.

## B.7 AgentOccam's Workflow Regarding the Planning Tree

0. At the beginning, we used the task objective as the root plan:

```
[0] (Active Plan) Find the solution to "Open my latest updated issue that has keyword "feature
    " in its title to check if it is closed"
```

1. The agent added a subplan to planning node 0 by issuing `branch [0] [Navigate to the Issues page to search for the latest issue with the keyword "feature" in the title.]`. Now the planning tree changed into:

```
[0] Find the solution to "Open my latest updated issue that has keyword "feature" in its title
    to check if it is closed"
    [1] (Active Plan) Navigate to the Issues page to search for the latest issue with the
        keyword "feature" in the title.
```

2. The agent navigated to the project's issue page.

3. The agent decomposed the plan by generating branch actions `branch [1] [Search for the latest issue with the keyword "feature" in the title and check if it is closed.]` and `branch [1] [Open the latest issue with the keyword "feature" in the title.]`. It performed navigation steps for each active plan before the next planning command was executed. In this example, after `branch [1] [Search for the latest issue with the keyword "feature" in the title and check if it is closed.]` was proposed, the active plan turned into "Search for the latest issue with the keyword 'feature' in the title and check if it is closed." The agent then typed the keyword "feature" into the search box and sorted the issues by operating the sort icon before generating the next plan "Open the latest issue with the keyword 'feature' in the title." In other words, all the navigation commands (e.g., search and sort) it issued were intended for the current active plan (e.g., search for the latest issue with the keyword "feature" in the title and check if it is closed). Just like a function call only needs to consider the function's scope, this allows the agent to only attend to the navigation actions dedicated to the active plan and the corresponding web observation as the playing history, which helps selectively replay past pages for the agent. Note that in this case, it assigned the two new sub plans to the same parent plan [1], which automatically shaped the planning tree's structure. Finally, the planning tree reformed into (the content enclosed in "[]" means comments, which is intended for illustration and didn't appear in the agent's prompt):

```
[0] Find the solution to "Open my latest updated issue that has keyword "feature" in its title
    to check if it is closed"
    [1] Navigate to the Issues page to search for the latest issue with the keyword "
        feature" in the title. [Plan's action scope: navigate.]
        [2] Search for the latest issue with the keyword "feature" in the title and
            check if it is closed. [Plan's action scope: search and sort.]
        [3] (Active Plan) Open the latest issue with the keyword "feature" in the
            title.
```

4. The agent executed the following steps to complete the task.

## C  AGENTOCCAM'S GENERALIZABILITY TO REAL-WORLD WEBSITES

Table 9: Performance comparison between Agent-E success rate (SR) and AgentOccam SR on tasks defined on real-world web environments from WebVoyager (He et al., 2024).

| Website (Task Number) | Agent-E SR | AgentOccam SR |
|---|---|---|
| Allrecipes (4) | **75.00%** | 50.00% |
| Amazon (1) | 0.00% | 0.00% |
| Apple (7) | **57.14%** | 28.57% |
| ArXiv (16) | **50.00%** | 43.75% |
| BBC News (2) | 0.00% | 0.00% |
| Booking (2) | 50.00% | **100.00%** |
| Cambridge Dictionary (9) | 55.56% | **88.89%** |
| Coursera (2) | 50.00% | 50.00% |
| ESPN (10) | 40.00% | **50.00%** |
| Google Map (9) | 33.33% | **44.44%** |
| Google Search (16) | 62.50% | **81.25%** |
| Huggingface (17) | 17.65% | **29.41%** |
| Wolfram Alpha (34) | **73.53%** | 61.76% |
| Overall (129) | 51.90% | **54.30%** |

WebVoyager benchmark (He et al., 2024) compiles web tasks based on 15 popular real-world websites. It comprises tasks with two types of user questions: ones with "golden" answers that are

Table 10: The success rate (SR) of AGENTOCCAM's ablation study with models GPT-4-TURBO and GEMINI-1.5-FLASH on the WebArena's development set.

| Agent | Model | SR (%) (#190) | Shopping (#48) | Shopping Admin (#41) | GitLab (#41) | Map (#29) | Reddit (#21) | Multisite (#10) |
|---|---|---|---|---|---|---|---|---|
| Vanilla | GPT-4-Turbo | 14.2 | 16.7 | 12.2 | 14.6 | 17.2 | 9.5 | **10.0** |
| ↓ Actions | GPT-4-Turbo | 25.8 | 22.9 | 24.4 | 34.2 | 31.0 | 19.1 | **10.0** |
| Above + X Scrolling | GPT-4-Turbo | 30.0 | 29.2 | 29.3 | 36.6 | 24.1 | 38.1 | **10.0** |
| Above + Obs Opt. | GPT-4-Turbo | 34.7 | 37.5 | 34.2 | 22.0 | 41.4 | **57.1** | **10.0** |
| Above + History | GPT-4-Turbo | 36.8 | 39.6 | 34.2 | 36.6 | **44.8** | 38.1 | **10.0** |
| AGENTOCCAM | GPT-4-Turbo | **44.2** | **45.8** | **46.3** | **48.8** | 41.4 | 47.6 | **10.0** |
| Vanilla | Gemini-1.5-Flash | 11.6 | 20.8 | 4.9 | 9.8 | 17.2 | 4.8 | 0.0 |
| ↓ Actions | Gemini-1.5-Flash | 23.2 | 29.2 | 22.0 | 29.3 | 13.8 | 19.1 | **10.0** |
| Above + X Scrolling | Gemini-1.5-Flash | 24.2 | 33.3 | 24.4 | 22.0 | 27.6 | 14.3 | 0.0 |
| Above + Obs Opt. | Gemini-1.5-Flash | 30.0 | **37.5** | **34.2** | 31.7 | 20.7 | 23.8 | **10.0** |
| Above + History | Gemini-1.5-Flash | 32.1 | 35.4 | 31.7 | 31.7 | **37.9** | **28.6** | **10.0** |
| AGENTOCCAM | Gemini-1.5-Flash | **33.7** | **37.5** | **34.2** | **36.6** | **37.9** | **28.6** | 0.0 |

definite and time-invariant, and ones with "possible" answers that are either open-ended with multiple potential answers or related to real-time information. We use WebVoyager questions with golden answers to avoid subjective human evaluations. We acknowledge that other WebVoyager tasks might test more web agent skills, but since they are not defined on static web pages, and thus their solutions would change over time, each new web agent's success rate would require human assessment of trajectories. Due to the subjectivity of human annotation (as evidenced by the high variance in our main experiment's human annotations provided alongside the code[8]), these results aren't comparable. We can't expect the same group of human annotators to label all previous agents' trajectories for every new agent, so we regretfully omit other tasks. Additionally, we exclude GitHub tasks, as the site's anti-scraping measures frequently cause page loading timeouts. Additionally, due to these measures, GitHub limits interactable elements, which prevents web page proper functionality, and the IP address hosting the agent is at risk of being banned. In contrast, WebArena's simulated GitLab environment mimics GitHub, enabling us to demonstrate AGENTOCCAM's performance on similar tasks using existing results. In a nutshell, we have 129 tasks from WebVoyager with definite golden answers across 13 real-world web environments, including Allrecipes, Amazon, Apple, ArXiv, BBC News, Booking, Cambridge Dictionary, Coursera, ESPN, Google Map, Google Search, Huggingface, and Wolfram Alpha.

Our baseline on WebVoyager is the concurrent work Agent-E (Abuelsaad et al., 2024), which introduces several architectural improvements like the planner-browser-reflector agent architecture and the browser's document object model selection. It achieves the previous SOTA on the full WebVoyager with the assessments done by humans. As they didn't report agent trajectory logs, we replicate their work on the WebVoyager subset introduced in the above paragraph, evaluated with the same definite-answer-based hard-coded evaluators as ours. We run each task once for both agents.

Based on the results in Table 9, AGENTOCCAM performs better than Agent-E on WebVoyager's definite-answer-subset. Specifically, both agents have their specificities. We find that Agent-E makes an edge on websites like Wolfram Alpha with more delicate plans issued by its "planner," and AGENTOCCAM outperforms it on websites like Google Search and Hugging Face with more accurate task interpretation and information retrieval. The impressive results across 10+ websites and many types of tasks can show AGENTOCCAM's generalizability in the web environment.

# D AGENTOCCAM'S GENERALIZABILITY TO BASE MODEL FAMILIES

We conduct the full set of ablation studies on a WebArena development subset with GEMINI-1.5-FLASH (Google, 2024), a model trained with different data from the GPT model family.

Due to the cost constraints, we construct a representative subset from the original 812 tasks in WebArena. Specifically, we sample one task from each task cluster instantiated with the same intent

---

[8]We would like to thank Yuhao Cheng, Tong Li, Yuren Hao, Ye Wu, and Xiaoxuan Wang for helping assess the agent trajectories.

Table 11: Action statistics for the ablation study of AGENTOCCAM's components with models GPT-4-TURBO and GEMINI-1.5-FLASH on the WebArena's development set.

| Exp. | Model | click | hover | type | press | scroll | new_tab | go_back | goto | note | stop | go_home | branch | prune |
|---|---|---|---|---|---|---|---|---|---|---|---|---|---|---|
| Vanilla | GPT-4-Turbo | 473 | 26 | 212 | 1 | 22 | 4 | 9 | 115 | - | - | - | - | - |
| ↓ Actions | GPT-4-Turbo | 1612 | - | 515 | - | 107 | - | 16 | - | 40 | 120 | 5 | - | - |
| Above + X Scrolling | GPT-4-Turbo | 1641 | - | 491 | - | - | - | 19 | - | 60 | 128 | 5 | - | - |
| Above + Obs Opt. | GPT-4-Turbo | 1539 | - | 437 | - | - | - | 21 | - | 82 | 137 | 1 | - | - |
| Above + History | GPT-4-Turbo | 1068 | - | 287 | - | - | - | 24 | - | 23 | 181 | 24 | - | - |
| AGENTOCCAM | GPT-4-Turbo | 1110 | - | 261 | - | - | - | 123 | 1 | 40 | 183 | 3 | 9 | 6 |
| Vanilla | Gemini-1.5-Flash | 103 | 0 | 65 | 0 | 7 | 5 | 0 | 9 | - | - | - | - | - |
| ↓ Actions | Gemini-1.5-Flash | 1390 | - | 509 | - | 176 | - | 28 | - | 31 | 130 | 11 | - | - |
| Above + X Scrolling | Gemini-1.5-Flash | 1258 | - | 542 | - | - | - | 23 | - | 53 | 134 | 22 | - | - |
| Above + Obs Opt. | Gemini-1.5-Flash | 1322 | - | 377 | - | - | - | 28 | - | 29 | 144 | 42 | - | - |
| Above + History | Gemini-1.5-Flash | 776 | - | 253 | - | - | - | 50 | - | 44 | 185 | 15 | - | - |
| AGENTOCCAM | Gemini-1.5-Flash | 942 | - | 289 | - | - | - | 67 | - | 44 | 185 | 34 | 27 | 75 |

Table 12: Average observation tokens per step of AGENTOCCAM's components with models GPT-4-TURBO and GEMINI-1.5-FLASH on the WebArena's development set. We use the GPT2 tokenizer from HUGGINGFACE (Radford et al., 2019).

| Exp. | Model | All | Shopping | Shopping Admin | GitLab | Map | Reddit | Multisite |
|---|---|---|---|---|---|---|---|---|
| Vanilla | GPT-4-Turbo | 2202.5 | 2268.7 | 2421.6 | 2267.3 | 1882.3 | 2119.0 | 1715.8 |
| ↓ Actions | GPT-4-Turbo | 1682.1 | 1496.7 | 2186.6 | 2168.3 | 953.3 | 1102.3 | 1261.8 |
| Above + X Scrolling | GPT-4-Turbo | 3571.2 | 3120.3 | 5175.5 | 4234.9 | 1423.3 | 3079.8 | 1977.9 |
| Above + Obs Opt. | GPT-4-Turbo | 2669.8 | 1664.9 | 4274.6 | 2744.3 | 1930.8 | 2664.4 | 1370.6 |
| Above + History | GPT-4-Turbo | 3107.0 | 1745.4 | 4988.5 | 3579.2 | 788.0 | 2758.4 | 1856.9 |
| AGENTOCCAM | GPT-4-Turbo | 2785.1 | 1500.8 | 5032.8 | 3032.3 | 645.6 | 3476.0 | 1233.9 |
| Vanilla | Gemini-1.5-Flash | 2303.4 | 2307.0 | 2537.5 | 2707.0 | 1849.7 | 1794.7 | 2037.6 |
| ↓ Actions | Gemini-1.5-Flash | 1713.9 | 1577.4 | 2112.3 | 2257.1 | 818.5 | 1542.8 | 1177.2 |
| Above + X Scrolling | Gemini-1.5-Flash | 3219.6 | 3110.7 | 5234.0 | 3378.0 | 1040.7 | 3204.9 | 2021.1 |
| Above + Obs Opt. | Gemini-1.5-Flash | 2814.9 | 1769.8 | 4632.9 | 3410.7 | 858.7 | 3203.5 | 1577.0 |
| Above + History | Gemini-1.5-Flash | 3283.2 | 1720.7 | 5078.3 | 3657.6 | 734.2 | 5712.5 | 1112.4 |
| AGENTOCCAM | Gemini-1.5-Flash | 2872.2 | 1639.6 | 4156.3 | 2519.3 | 688.7 | 6241.4 | 1267.7 |

template (e.g., "What is the top-{{n}} best-selling product in {{year}}", where "n" and "{{year}}" would be replaced by instantiation tokens) in WebArena, forming a development set with 190 tasks[9]. We use Gemini-1.5-flash for all experiments. We run each task once and restart the experiment if the WebArena simulator fails (i.e., login expires, Reddit post limit exceeds, map malfunctioning, etc.). The results and statistics are listed in Table 10, 11, 12, 13, with the performance from the GPT-4-turbo counterpart on the same development task set for comparison.

We can observe a similar trend with the Gemini model that each alignment component introduced by our paper contributes to the web agent's overall performance. To be specific, removing non-essential actions (↓ Actions) encourages the agent to explore more actively with commands `click` and `type`; disabling scrolling (Above + X Scrolling) proves advantageous in tasks where key information is not presented on the first browser sight; simplifying web page elements (Above + Obs Opt.) reduces the observation token number; selectively replaying web element in one page (Above + History) reduces the steps required to accomplish the task; and planning and selectively replay past pages (AGENTOCCAM) enables the agent to self-organize task workflow and quickly restore to a previous stage after several failed attempts. In conclusion, each alignment component proposed by AGENTOCCAM proves beneficial to the agent system and can be generalized to different base models.

---

[9]We remove `stop` in the statistics for the vanilla WebArena agent as this action is excluded in their officially defined action space. However, their agent is allowed by code to generate `stop` to end the trajectory.

[9]The task ids are: 3, 9, 13, 20, 22, 30, 35, 40, 42, 44, 46, 48, 52, 61, 65, 68, 71, 76, 78, 80, 86, 89, 94, 96, 97, 99, 104, 109, 116, 117, 118, 120, 125, 127, 131, 132, 138, 141, 148, 153, 156, 157, 158, 164, 171, 173, 180, 187, 188, 194, 204, 205, 212, 217, 219, 224, 225, 230, 234, 237, 239, 244, 250, 257, 258, 259, 262, 265, 271, 274, 281, 283, 285, 287, 289, 294, 298, 305, 311, 313, 316, 323, 324, 333, 337, 340, 345, 350, 354, 356, 357, 360, 367, 368, 370, 375, 376, 377, 382, 383, 384, 386, 388, 390, 395, 401, 406, 410, 413, 416, 419, 423, 425, 435, 440, 443, 446, 452, 457, 463, 468, 472, 476, 482, 490, 495, 499, 503, 508, 509, 512, 518, 521, 525, 530, 535, 539, 546, 550, 555, 561, 566, 567, 571, 578, 580, 587, 594, 599, 604, 606, 613, 615, 624, 627, 634, 637, 644, 646, 653, 660, 663, 666, 669, 674, 677, 682, 687, 691, 698, 702, 706, 713, 715, 720, 730, 731, 738, 749, 754, 758, 762, 766, 770, 771, 772, 779, 785, 797, 803.

Table 13: Average number of steps per task of AGENTOCCAM's components with models GPT-4-TURBO and GEMINI-1.5-FLASH on the WebArena's development set.

| Exp. | Model | All | Shopping | Shopping Admin | GitLab | Map | Reddit | Multisite |
|---|---|---|---|---|---|---|---|---|
| Vanilla | GPT-4-Turbo | 1.3 | 1.5 | 1.3 | 1.1 | 1.4 | 1.3 | 0.9 |
| ↓ Actions | GPT-4-Turbo | 3.0 | 2.9 | 3.1 | 3.2 | 3.3 | 2.4 | 2.4 |
| Above + X Scrolling | GPT-4-Turbo | 2.9 | 2.2 | 3.3 | 3.2 | 3.5 | 2.2 | 2.8 |
| Above + Obs Opt. | GPT-4-Turbo | 2.7 | 2.4 | 2.7 | 3.4 | 2.7 | 2.4 | 2.7 |
| Above + History | GPT-4-Turbo | 2.0 | 1.3 | 2.4 | 2.4 | 1.8 | 1.4 | 3.2 |
| AGENTOCCAM | GPT-4-Turbo | 2.1 | 1.7 | 2.2 | 2.4 | 2.2 | 2.0 | 3.0 |
| Vanilla | Gemini-1.5-Flash | 0.5 | 0.5 | 0.4 | 0.5 | 0.5 | 0.3 | 0.6 |
| ↓ Actions | Gemini-1.5-Flash | 2.8 | 3.2 | 2.8 | 2.5 | 2.6 | 2.9 | 2.4 |
| Above + X Scrolling | Gemini-1.5-Flash | 2.5 | 2.4 | 2.2 | 2.4 | 2.7 | 2.7 | 3.3 |
| Above + Obs Opt. | Gemini-1.5-Flash | 2.4 | 2.5 | 2.3 | 2.8 | 2.4 | 1.9 | 2.2 |
| Above + History | Gemini-1.5-Flash | 1.6 | 1.6 | 1.7 | 1.7 | 1.4 | 1.5 | 1.7 |
| AGENTOCCAM | Gemini-1.5-Flash | 2.0 | 1.6 | 2.4 | 2.0 | 2.2 | 1.8 | 2.8 |

# E  EVALUATOR RECTIFICATIONS

## E.1  RECTIFICATION CATEGORIZATION

We only modify the evaluator when it's deemed erroneous due to the wrong task labels or misuse of evaluating functions. When the task definition and corresponding evaluation metric match to some extent but might be misleading to most agents and even to human, we still keep the original ones to ensure the slightest reasonable changes. **We emphasize that we re-implement WebArena's base agent SteP's agent with the same web environment and modified evaluators as AGENTOCCAM for a fair comparison.** For example, we keep the evaluators of shopping tasks defined with template 163, requiring the agent to `"Draft an email to the shop owner via their contact us function for a coupon as {reason}"`, which doesn't explicitly specify whether to submit the drafted email. However, the evaluator is defined to assess the not yet submitted email. All capable LLM-based agents we have tested, which have been aligned to be helpful, will for sure submit the email if not directly prompted to behave in the way the evaluator desires, leading the email field to be blank and thus failing those tasks. Another example of this kind is the Reddit task asking the agent to find the most appropriate subreddit to post (task template 6100), where the assessment of appropriation is very subjective. In all those tasks, we follow the original evaluators, though their evaluation outcomes are arguably questionable.

We categorize our evaluator modifications into two classes, namely label errors and improper evaluation function selection, raise representative examples for each class, and list all the changes made.[10]

**Label errors**: We find there exist evaluator definition errors and some typos in the correct answers. In the later cases, the tasks always require exact matching, where any well-aligned LLM-agent would correct those typos in their generation. We thus rectify those errors:

*i)* Evaluator definitions contain errors. For example, in the Reddit task 584, the evaluator would open up the wrong page for the evaluation. Another case in point is the shopping task 261, where the `url_match` evaluator is constrained to identifying one correct url (`<server_host>:7770/electronics/headphones.html`), misjudging the same page (of the identical content) with a different url (`<server_host>:7770/electronics.html?cat=60`). Tasks fall in this category include: **261-264, 351-355, 584, 644, 679, 707-709, 786**.

*ii)* The answer contains typos or grammatical errors. For example the `is car necessary in NYC` in task 601, or the `budge` in task 603. More tasks of this kind include: **task id 240, 298-302, 489, 583, 601, 603, 606, 608, 629, 645-649**.

**Improper evaluation function selection**: Evaluator problems are more obvious in this case with the following types:

---

[10]As the evaluator is programmed by the WebArena simulator to be revoked only once at the end of each trajectory, our statement of "our approach outperforms the baseline methods with the original evaluators" refers to setting all the rewards of the trajectories with modified evaluators to be 0, which can be verified with the reported trajectory logs.

*i)* Use the `exact_match` function that compares whether the answer given by a human label-er and the answer returned by the agent is identically the same. Errors occur when the agent returns a full-form or a more complete answer, where the evaluators' labels cannot match. For example, in Reddit task 644 that requires the agent to post a meeting notice with the meeting date, where the keyword match for such date is exactly the `Dec 15th`, where the evaluator would judge other answers like `December 15th` as incorrect, where we change the keyword matching to one that could match both `Dec 15th` and `December 15th`. (In other cases with a single answer, we simply replace `exact_match` with `fuzzy_match`, which for instance in task 254 it could match `4125785000` with the agent's answer `The phone number is 4125785000`; or replace `exact_match` with `must_include`, which for instance in task 363 it could match `778m` with the agent's answer `778 m`.) It also demands that the answer should strictly include expressions like `virtual meetup`, where the agent might add other words in the `virtual` and `meetup`. In that sense, we also split the keyword `virtual meetup` into two separate keywords, i.e., `virtual` and `meetup`. Tasks of this kind include: **task id 97, 118, 146, 178-182, 254, 293-297, 308-312, 330, 363-367, 415-418, 528-532, 640-649, 653-657**.

*ii)* Use the poorly defined `fuzzy_match` function, that would view the answer returned as un-qualified for the missing-from-expression answer exploration process, or assess answers with more detailed answers as partially correct (reward=0). We thus shift our prompt for the `fuzzy_match` function from: *'Help a teacher to grade the answer of a student given a question. Keep in mind that the student may use different phrasing or wording to answer the question. The goal is to evaluate whether the answer is semantically equivalent to the reference answer.'* to *'Help a teacher to grade the answer of a student given a question. **Keep in mind that the student has executed the actions to get the answer.** They are allowed to use different phrasing or wording to answer the question. The goal is to evaluate whether the key points in the reference answer are included in the student's answer. **We allow answers with additional information that doesn't contradict the reference answer and review them as fully (not partially) correct.''*

*iii)* Misuse the `fuzzy_match` function by splitting the keyword list for matching into a list, where each of the keyword and the entire answer, would be evaluated as partially correct (reward=0). In other words, no answer would be assessed as the correct answer (even the gloden-standard answer itself) due to such evaluator function misuse. This could be inferred from the function and the evaluator's definition. Tasks of this type include: **task id 16-20,** In such tasks, we simply merge the list of keywords into a string, concatenated with `"; "`. For instance, for task 16, the previous `fuzzy_match` field is `["driving: 2min", "walking: 16min"]`, and we modify it to `["driving: 2min; walking: 16min"]`.

## E.2 DETAILS

```
# Task 16
### eval.reference_answers.fuzzy_match
['driving: 2min', 'walking: 16min']
['driving: 2min; walking: 16min']

# Task 17
### eval.reference_answers.fuzzy_match
['driving: 13min', 'walking: 1h 35min']
['driving: 13min; walking: 1h 35min']

# Task 18
### eval.reference_answers.fuzzy_match
['driving: 15min', 'walking: 1h 47min']
['driving: 15min; walking: 1h 47min']

# Task 19
### eval.reference_answers.fuzzy_match
['driving: 12min', 'walking: 1h 44min.']
['driving: 12min; walking: 1h 44min.']

# Task 20
### eval.reference_answers.fuzzy_match
['driving: 13min', 'walking: 1h 45min']
['driving: 13min; walking: 1h 45min']

# Task 97
### eval.reference_answers.must_include
['914km']
['914km |OR| 914 km']

# Task 118
### eval.program_html
[{'url': 'last', 'locator': '', 'required_contents': {'must_include': ['jaw bruxism', 'mouth guard']}}]
```

```
[{'url': 'last', 'locator': '', 'required_contents': {'must_include': ['jaw', 'bruxism', 'mouth guard']}}]

# Task 146
### eval.reference_answers.must_include
['16x24']
### eval.reference_answers.fuzzy_match
['16x24']

# Task 178
### eval.reference_answers.exact_match
Yes
### eval.reference_answers.fuzzy_match
['Yes, it is closed']

# Task 179
### eval.reference_answers.exact_match
Yes
### eval.reference_answers.fuzzy_match
['Yes, it is closed']

# Task 180
### eval.reference_answers.exact_match
No
### eval.reference_answers.fuzzy_match
['No, it is open']

# Task 181
### eval.reference_answers.exact_match
No
### eval.reference_answers.fuzzy_match
['No, it is open']

# Task 182
### eval.reference_answers.exact_match
Yes
### eval.reference_answers.fuzzy_match
['Yes, it is closed']

# Task 240
### instantiation_dict.product_category
competitive swimwear
competative swimwear
### intent
I am doing a market survey for one stop market, show me the most expensive product from competitive swimwear
        category
I am doing a market survey for one stop market, show me the most expensive product from competative swimwear
        category

# Task 254
### eval.reference_answers.exact_match
4125785000
### eval.reference_answers.fuzzy_match
['4125785000']

# Task 261
### eval.or
[{'reference_url': 'http://localhost:7770/electronics.html?cat=60'}]

# Task 262
### eval.or
[{'reference_url': 'http://localhost:7770/clothing-shoes-jewelry.html?cat=145'}]

# Task 263
### eval.or
[{'reference_url': 'http://localhost:7770/clothing-shoes-jewelry.html?cat=143'}]

# Task 264
### eval.or
[{'reference_url': 'http://localhost:7770/office-products.html?cat=187'}]

# Task 293
### eval.reference_answers.exact_match
git clone ssh://git@metis.lti.cs.cmu.edu:2222/convexegg/super_awesome_robot.git
### eval.reference_answers.must_include
['git clone ssh://git@metis.lti.cs.cmu.edu:2222/convexegg/super_awesome_robot.git']

# Task 294
### eval.reference_answers.exact_match
git clone ssh://git@metis.lti.cs.cmu.edu:2222/convexegg/chatgpt.git
### eval.reference_answers.must_include
['git clone ssh://git@metis.lti.cs.cmu.edu:2222/convexegg/chatgpt.git']

# Task 295
### eval.reference_answers.exact_match
git clone ssh://git@metis.lti.cs.cmu.edu:2222/root/metaseq.git
### eval.reference_answers.must_include
['git clone ssh://git@metis.lti.cs.cmu.edu:2222/root/metaseq.git']

# Task 296
### eval.reference_answers.exact_match
ssh://git@metis.lti.cs.cmu.edu:2222/eriklindernoren/PyTorch-GAN.git
### eval.reference_answers.must_include
```

```
['git clone ssh://git@metis.lti.cs.cmu.edu:2222/eriklindernoren/PyTorch-GAN.git']

# Task 297
### eval.reference_answers.exact_match
ssh://git@metis.lti.cs.cmu.edu:2222/yjlou/2019-nCov.git
### eval.reference_answers.must_include
['git clone ssh://git@metis.lti.cs.cmu.edu:2222/yjlou/2019-nCov.git']

# Task 298
### intent_template
Show the most recent {{status}} order
Show the most recent {{status}} order page
### intent
Show the most recent completed order
Show the most recent completed order page

# Task 299
### intent_template
Show the most recent {{status}} order
Show the most recent {{status}} order page
### intent
Show the most recent cancelled order
Show the most recent cancelled order page

# Task 300
### intent_template
Show the most recent {{status}} order
Show the most recent {{status}} order page
### intent
Show the most recent pending order
Show the most recent pending order page

# Task 301
### intent_template
Show the most recent {{status}} order
Show the most recent {{status}} order page
### intent
Show the most recent processing order
Show the most recent processing order page

# Task 302
### intent_template
Show the most recent {{status}} order
Show the most recent {{status}} order page
### intent
Show the most recent out of delivery order
Show the most recent out of delivery order page

# Task 308
### eval.reference_answers.exact_match
Shawn Allen
### eval.reference_answers.must_include
['Shawn Allen']

# Task 309
### eval.reference_answers.exact_match
Grayson Wright
### eval.reference_answers.must_include
['Grayson Wright']

# Task 310
### eval.reference_answers.exact_match
tokudu
### eval.reference_answers.must_include
['tokudu']

# Task 311
### eval.reference_answers.exact_match
Erik Linder-Nor\'en
### eval.reference_answers.must_include
['Erik Linder-Nor\'en']

# Task 312
### eval.reference_answers.exact_match
Christopher Groskopf
### eval.reference_answers.must_include
['Christopher Groskopf']

# Task 330
### eval.reference_answers.must_include
['81.31']
['83.31']
### eval.reference_answer_raw_annotation
81.31
83.31

# Task 351
### eval.or
[{'reference_url': 'http://localhost:7770/video-games.html?cat=67&product_list_order=price'}]

# Task 352
### eval.or
```

```
[{'reference_url': 'http://localhost:7770/health-household.html?cat=192&product_list_order=price'}]

# Task 353
### instantiation_dict.product_category
competitive swimwear
competative swimwear
### intent
List products from competitive swimwear category by ascending price
List products from competative swimwear category by ascending price
### eval.or
[{'reference_url': 'http://localhost:7770/clothing-shoes-jewelry.html?cat=149&product_list_order=price'}]

# Task 354
### eval.or
[{'reference_url': 'http://localhost:7770/home-kitchen.html?cat=154&product_list_order=price&product_list_dir=
    desc'}]

# Task 355
### eval.or
[{'reference_url': 'http://localhost:7770/home-kitchen.html?cat=155&product_list_dir=desc'}]

# Task 363
### eval.reference_answers.exact_match
748m
### eval.reference_answers.must_include
['778m |OR| 778 m']
### eval.reference_answer_raw_annotation
748m
778m

# Task 364
### eval.reference_answers.exact_match
1.7km
### eval.reference_answers.must_include
['1.7km |OR| 1.7 km']

# Task 365
### eval.reference_answers.exact_match
2.2km
### eval.reference_answers.must_include
['2.2km |OR| 2.2 km']

# Task 366
### eval.reference_answers.exact_match
1.2km
### eval.reference_answers.must_include
['1.2km |OR| 1.2 km']

# Task 367
### eval.reference_answers.exact_match
1.4km
1.4km |OR| 1.4 km

# Task 415
### eval.program_html
[{'url': 'http://localhost:8023/byteblaze/a11y-webring.club/-/merge_requests/40', 'locator': 'document.
    querySelector(\'[id="notes-list"]\').lastElementChild.querySelector(\'.timeline-discussion-body\').
    outerText', 'required_contents': {'exact_match': '@davepgreene'}}]
[{'url': 'http://localhost:8023/byteblaze/a11y-webring.club/-/merge_requests/40', 'locator': 'document.
    querySelector(\'[id="notes-list"]\').lastElementChild.querySelector(\'.timeline-discussion-body\').
    outerText', 'required_contents': {'must_include': ['@davepgreene']}}]

# Task 416
### eval.program_html
[{'url': 'http://localhost:8023/a11yproject/a11yproject.com/-/merge_requests/1270', 'locator': 'document.
    querySelector(\'[id="notes-list"]\').lastElementChild.querySelector(\'.timeline-discussion-body\').
    outerText', 'required_contents': {'exact_match': 'Thank you'}}]
[{'url': 'http://localhost:8023/a11yproject/a11yproject.com/-/merge_requests/1270', 'locator': 'document.
    querySelector(\'[id="notes-list"]\').lastElementChild.querySelector(\'.timeline-discussion-body\').
    outerText', 'required_contents': {'must_include': ['Thank you']}}]

# Task 417
### eval.program_html
[{'url': 'http://localhost:8023/a11yproject/a11yproject.com/-/merge_requests/1485', 'locator': 'document.
    querySelector(\'[id="notes-list"]\').lastElementChild.querySelector(\'.timeline-discussion-body\').
    outerText', 'required_contents': {'exact_match': '@Roshanjossey'}}]
[{'url': 'http://localhost:8023/a11yproject/a11yproject.com/-/merge_requests/1485', 'locator': 'document.
    querySelector(\'[id="notes-list"]\').lastElementChild.querySelector(\'.timeline-discussion-body\').
    outerText', 'required_contents': {'must_include': ['@Roshanjossey']}}]

# Task 418
### eval.program_html
[{'url': 'http://localhost:8023/byteblaze', 'locator': "document.querySelector('.cover-status').lastChild.
    textContent", 'required_contents': {'exact_match': 'Busy'}}]
[{'url': 'http://localhost:8023/byteblaze', 'locator': "document.querySelector('.cover-title').textContent", '
    required_contents': {'must_include': ['Busy']}}]

# Task 489
### instantiation_dict.heading
No privacy policy is needed is this dystopian world
No privacy policy is needed in this dystopian world
### intent
```

Change the page title of "Privacy Policy" page on my site to "No privacy policy is needed is this dystopian world".
Change the page title of "Privacy Policy" page on my site to "No privacy policy is needed in this dystopian world".
### eval.program_html
[{'url': 'http://localhost:7780/admin/cms/page/edit/page_id/4/', 'locator': 'document.querySelector(\'input[name="title"\').value', 'required_contents': {'exact_match': 'No privacy policy is needed is this dystopian world'}}]
[{'url': 'http://localhost:7780/admin/cms/page/edit/page_id/4/', 'locator': 'document.querySelector(\'input[name="title"\').value', 'required_contents': {'exact_match': 'No privacy policy is needed in this dystopian world'}}]

# Task 528
### eval.program_html
[{'url': 'last', 'locator': 'document.querySelector(\'[title="What's on your mind?"\').value', 'required_contents': {'must_include': ['refund', 'it broke after three days of use', '000000180', '12.99']}}]
[{'url': 'last', 'locator': 'document.querySelector(\'[title="What's on your mind?"\').value', 'required_contents': {'must_include': ['refund', 'broke', 'three days of use', '000000180', '12.99']}}]

# Task 529
### eval.program_html
[{'url': 'last', 'locator': 'document.querySelector(\'[title="What's on your mind?"\').value', 'required_contents': {'must_include': ['refund', 'it broke after three days of use', '000000148', '169.95']}}]
[{'url': 'last', 'locator': 'document.querySelector(\'[title="What's on your mind?"\').value', 'required_contents': {'must_include': ['refund', 'broke', 'three days of use', '000000148', '169.95']}}]

# Task 530
### eval.program_html
[{'url': 'last', 'locator': 'document.querySelector(\'[title="What's on your mind?"\').value', 'required_contents': {'must_include': ['refund', 'it broke after three days of use', '000000161', '68.88']}}]
[{'url': 'last', 'locator': 'document.querySelector(\'[title="What's on your mind?"\').value', 'required_contents': {'must_include': ['refund', 'broke', 'three days of use', '000000161', '68.88']}}]

# Task 531
### eval.program_html
[{'url': 'last', 'locator': 'document.querySelector(\'[title="What's on your mind?"\').value', 'required_contents': {'must_include': ['refund', 'it broke after three days of use', '000000180', '$12.99']}}]
[{'url': 'last', 'locator': 'document.querySelector(\'[title="What's on your mind?"\').value', 'required_contents': {'must_include': ['refund', 'broke', 'three days of use', '000000180', '$12.99']}}]

# Task 532
### eval.program_html
[{'url': 'last', 'locator': 'document.querySelector(\'[title="What's on your mind?"\').value', 'required_contents': {'must_include': ['refund', 'it broke after three days of use', '000000180', '1.63']}}]
[{'url': 'last', 'locator': 'document.querySelector(\'[title="What's on your mind?"\').value', 'required_contents': {'must_include': ['refund', 'broke', 'three days of use', '000000180', '1.63']}}]

# Task 583
### eval.program_html
[{'url': 'http://localhost:9999/f/PlantsForCatParents/edit', 'locator': 'document.querySelector("#forum_description").value', 'required_contents': {'must_include': ['Cat parents & plan lovers']}}, {'url': 'http://localhost:9999/f/PlantsForCatParents/edit', 'locator': 'document.querySelector("#forum_sidebar").value', 'required_contents': {'must_include': ['Cat friendly', 'Local vendors', 'Promotion', 'Toxic plants!']}}]
[{'url': 'http://localhost:9999/f/PlantsForCatParents/edit', 'locator': 'document.querySelector("#forum_description").value', 'required_contents': {'must_include': ['Cat parents & plant lovers']}}, {'url': 'http://localhost:9999/f/PlantsForCatParents/edit', 'locator': 'document.querySelector("#forum_sidebar").value', 'required_contents': {'must_include': ['Cat friendly', 'Local vendors', 'Promotion', 'Toxic plants!']}}]

# Task 584
### eval.program_html
[{'url': 'http://localhost:9999/f/Karaoke', 'locator': 'document.querySelector("#forum_description").value', 'required_contents': {'must_include': ['Place for Karaoke lovers']}}, {'url': 'http://localhost:9999/f/Karaoke', 'locator': 'document.querySelector("#forum_sidebar").value', 'required_contents': {'must_include': ['devices', 'setup']}}]
[{'url': 'http://localhost:9999/f/Karaoke/edit', 'locator': 'document.querySelector("#forum_description").value', 'required_contents': {'must_include': ['Place for Karaoke lovers']}}, {'url': 'http://localhost:9999/f/Karaoke/edit', 'locator': 'document.querySelector("#forum_sidebar").value', 'required_contents': {'must_include': ['devices', 'setup']}}]

# Task 601
### eval.program_html
[{'url': "func:reddit_get_post_url('__last_url__')", 'locator': "document.querySelector('.submission__inner').outerText", 'required_contents': {'must_include': ['is car necessary in NYC']}}]
[{'url': "func:reddit_get_post_url('__last_url__')", 'locator': "document.querySelector('.submission__inner').outerText", 'required_contents': {'must_include': ['is car necessary in NYC |OR| is a car necessary in NYC']}}]

# Task 603
### instantiation_dict.question
safe and budge apartment to live in nyc
safe and budget apartment to live in nyc
### intent
Post my question, "safe and budge apartment to live in nyc", in a subreddit where I'm likely to get an answer
Post my question, "safe and budget apartment to live in nyc", in a subreddit where I'm likely to get an answer
### eval.program_html

```
[{'url': "func:reddit_get_post_url('__last_url__')", 'locator': "document.querySelector('.submission__inner').
     outerText", 'required_contents': {'must_include': ['safe and budge apartment to live in nyc']}}]
[{'url': "func:reddit_get_post_url('__last_url__')", 'locator': "document.querySelector('.submission__inner').
     outerText", 'required_contents': {'must_include': ['safe and budget apartment to live in nyc']}}]
```

# Task 606
### eval.program_html
```
[{'url': "func:reddit_get_post_url('__last_url__')", 'locator': "document.querySelector('.submission__inner').
     outerText", 'required_contents': {'must_include': ['is car necessary']}}]
[{'url': "func:reddit_get_post_url('__last_url__')", 'locator': "document.querySelector('.submission__inner').
     outerText", 'required_contents': {'must_include': ['is car necessary |OR| is a car necessary']}}]
```

# Task 608
### instantiation_dict.question
safe and budge apartment to live
safe and budget apartment to live
### intent
Find a subreddit focused on topics related to city lives in DMV area, and post my question, "safe and budge
     apartment to live" there
Find a subreddit focused on topics related to city lives in DMV area, and post my question, "safe and budget
     apartment to live" there
### eval.program_html
```
[{'url': "func:reddit_get_post_url('__last_url__')", 'locator': "document.querySelector('.submission__inner').
     outerText", 'required_contents': {'must_include': ['safe and budge apartment to live']}}]
[{'url': "func:reddit_get_post_url('__last_url__')", 'locator': "document.querySelector('.submission__inner').
     outerText", 'required_contents': {'must_include': ['safe and budget apartment to live']}}]
```

# Task 629
### eval.program_html
```
[{'url': "func:reddit_get_post_url('__last_url__')", 'locator': "document.querySelector('.submission__inner').
     outerText", 'required_contents': {'must_include': ['your opinion', 'Fun thing to do in Pittsburgh']}}]
[{'url': "func:reddit_get_post_url('__last_url__')", 'locator': "document.querySelector('.submission__inner').
     outerText", 'required_contents': {'must_include': ['your opinion', 'Fun thing to do in Pittsburgh |OR|
     Fun things to do in Pittsburgh']}}]
```

# Task 640
### eval.program_html
```
[{'url': "func:reddit_get_post_url('__last_url__')", 'locator': "document.querySelector('.submission__inner').
     outerText", 'required_contents': {'must_include': ['book reading', 'March 15th', 'virtual meetup']}}]
[{'url': "func:reddit_get_post_url('__last_url__')", 'locator': "document.querySelector('.submission__inner').
     outerText", 'required_contents': {'must_include': ['book reading', 'Mar 15th |OR| March 15th', 'virtual
     ', 'meetup']}}]
```

# Task 641
### eval.program_html
```
[{'url': "func:reddit_get_post_url('__last_url__')", 'locator': "document.querySelector('.submission__inner').
     outerText", 'required_contents': {'must_include': ['Harry Potter', 'July 8th', 'virtual meetup']}}]
[{'url': "func:reddit_get_post_url('__last_url__')", 'locator': "document.querySelector('.submission__inner').
     outerText", 'required_contents': {'must_include': ['Harry Potter', 'Jul 8th |OR| July 8th', 'virtual', '
     meetup']}}]
```

# Task 642
### eval.program_html
```
[{'url': "func:reddit_get_post_url('__last_url__')", 'locator': "document.querySelector('.submission__inner').
     outerText", 'required_contents': {'must_include': ['Big little lies', 'Sep 10th', 'virtual meetup']}}]
[{'url': "func:reddit_get_post_url('__last_url__')", 'locator': "document.querySelector('.submission__inner').
     outerText", 'required_contents': {'must_include': ['Big little lies', 'Sep 10th |OR| September 10th', '
     virtual', 'meetup']}}]
```

# Task 643
### eval.program_html
```
[{'url': "func:reddit_get_post_url('__last_url__')", 'locator': "document.querySelector('.submission__inner').
     outerText", 'required_contents': {'must_include': ['racing cars', 'Oct 21st', 'virtual meetup']}}]
[{'url': "func:reddit_get_post_url('__last_url__')", 'locator': "document.querySelector('.submission__inner').
     outerText", 'required_contents': {'must_include': ['racing cars', 'Oct 21st |OR| October 21st', 'virtual
     ', 'meetup']}}]
```

# Task 644
### instantiation_dict.subreddit
games
gaming
### intent
Post a notice on a virtual meetup for Tears of Kingdom enthusiasts on Dec 15th in the games subreddit
Post a notice on a virtual meetup for Tears of Kingdom enthusiasts on Dec 15th in the gaming subreddit
### eval.reference_url
http://localhost:9999/f/games
http://localhost:9999/f/gaming
### eval.program_html
```
[{'url': "func:reddit_get_post_url('__last_url__')", 'locator': "document.querySelector('.submission__inner').
     outerText", 'required_contents': {'must_include': ['Tears of Kingdom', 'Dec 15th', 'virtual meetup']}}]
[{'url': "func:reddit_get_post_url('__last_url__')", 'locator': "document.querySelector('.submission__inner').
     outerText", 'required_contents': {'must_include': ['Tears of Kingdom', 'Dec 15th |OR| December 15th', '
     virtual', 'meetup']}}]
```

# Task 645
### intent_template
Post in {{subreddit}} subreddit about what could machine learning help the correpong field.
Post in {{subreddit}} subreddit about what could machine learning help the corresponding field.
### intent
Post in books subreddit about what could machine learning help the correpong field.
Post in books subreddit about what could machine learning help the corresponding field.
### eval.program_html

```
[{'url': "func:reddit_get_post_url('__last_url__')", 'locator': "document.querySelector('.submission__inner').
    outerText", 'required_contents': {'must_include': ['machine learning', 'help']}}]
[{'url': "func:reddit_get_post_url('__last_url__')", 'locator': "document.querySelector('.submission__inner').
    outerText", 'required_contents': {'must_include': ['machine learning', 'help |OR| assist |OR| benefit |
    OR| improve |OR| enhance |OR| transform |OR| revolutionize']}}]
```

# Task 646
### intent_template
Post in {{subreddit}} subreddit about what could midjourney help the correpong field.
Post in {{subreddit}} subreddit about what could midjourney help the corresponding field.
### intent
Post in DIY subreddit about what could midjourney help the correpong field.
Post in DIY subreddit about what could midjourney help the corresponding field.
### eval.program_html
```
[{'url': "func:reddit_get_post_url('__last_url__')", 'locator': "document.querySelector('.submission__inner').
    outerText", 'required_contents': {'must_include': ['midjourney', 'help']}}]
[{'url': "func:reddit_get_post_url('__last_url__')", 'locator': "document.querySelector('.submission__inner').
    outerText", 'required_contents': {'must_include': ['midjourney', 'help |OR| assist |OR| benefit |OR|
    improve |OR| enhance |OR| transform |OR| revolutionize']}}]
```

# Task 647
### intent_template
Post in {{subreddit}} forum about what could open-source LLMs help the correpong field.
Post in {{subreddit}} forum about what could open-source LLMs help the corresponding field.
### intent
Post in technology forum about what could open-source LLMs help the correpong field.
Post in technology forum about what could open-source LLMs help the corresponding field.
### eval.program_html
```
[{'url': "func:reddit_get_post_url('__last_url__')", 'locator': "document.querySelector('.submission__inner').
    outerText", 'required_contents': {'must_include': ['open-source LLMs', 'help']}}]
[{'url': "func:reddit_get_post_url('__last_url__')", 'locator': "document.querySelector('.submission__inner').
    outerText", 'required_contents': {'must_include': ['open-source LLMs', 'help |OR| assist |OR| benefit |
    OR| improve |OR| enhance |OR| transform |OR| revolutionize']}}]
```

# Task 648
### intent_template
Post in {{subreddit}} forum about what could large language models help the correpong field.
Post in {{subreddit}} forum about what could large language models help the corresponding field.
### intent
Post in dataisbeautiful forum about what could large language models help the correpong field.
Post in dataisbeautiful forum about what could large language models help the corresponding field.
### eval.program_html
```
[{'url': "func:reddit_get_post_url('__last_url__')", 'locator': "document.querySelector('.submission__inner').
    outerText", 'required_contents': {'must_include': ['large language models', 'help']}}]
[{'url': "func:reddit_get_post_url('__last_url__')", 'locator': "document.querySelector('.submission__inner').
    outerText", 'required_contents': {'must_include': ['large language models', 'help |OR| assist |OR|
    benefit |OR| improve |OR| enhance |OR| transform |OR| revolutionize']}}]
```

# Task 649
### intent_template
Post in {{subreddit}} subreddit about what could diffusion model help the correpong field.
Post in {{subreddit}} subreddit about what could diffusion model help the corresponding field.
### intent
Post in history subreddit about what could diffusion model help the correpong field.
Post in history subreddit about what could diffusion model help the corresponding field.
### eval.program_html
```
[{'url': "func:reddit_get_post_url('__last_url__')", 'locator': "document.querySelector('.submission__inner').
    outerText", 'required_contents': {'must_include': ['diffusion model', 'help']}}]
[{'url': "func:reddit_get_post_url('__last_url__')", 'locator': "document.querySelector('.submission__inner').
    outerText", 'required_contents': {'must_include': ['diffusion model', 'help |OR| assist |OR| benefit |OR
    | improve |OR| enhance |OR| transform |OR| revolutionize']}}]
```

# Task 653
### eval.program_html
```
[{'url': 'last', 'locator': 'document.querySelector(\'[title="What's on your mind?"\').value', '
    required_contents': {'must_include': ['refund', 'it broke after three days of use', '000000180', '
    B087QJN9W1']}}]
[{'url': 'last', 'locator': 'document.querySelector(\'[title="What's on your mind?"\').value', '
    required_contents': {'must_include': ['refund', 'broke after', 'three days of use', '000000180', '
    B087QJN9W1']}}]
```

# Task 654
### eval.program_html
```
[{'url': 'last', 'locator': 'document.querySelector(\'[title="What's on your mind?"\').value', '
    required_contents': {'must_include': ['refund', 'it broke after three days of use', '161', 'B09P7BFL4H
    ']}}]
[{'url': 'last', 'locator': 'document.querySelector(\'[title="What's on your mind?"\').value', '
    required_contents': {'must_include': ['refund', 'broke', 'three days of use', '161', 'B09P7BFL4H']}}]
```

# Task 655
### eval.program_html
```
[{'url': 'last', 'locator': 'document.querySelector(\'[title="What's on your mind?"\').value', '
    required_contents': {'must_include': ['refund', 'it broke after three days of use', '180', 'B087QJN9W1
    ']}}]
[{'url': 'last', 'locator': 'document.querySelector(\'[title="What's on your mind?"\').value', '
    required_contents': {'must_include': ['refund', 'broke', 'three days of use', '180', 'B087QJN9W1']}}]
```

# Task 656
### eval.program_html
```
[{'url': 'last', 'locator': 'document.querySelector(\'[title="What's on your mind?"\').value', '
    required_contents': {'must_include': ['refund', 'it broke after three days of use', '180', 'B0041MSF2S
    ']}}]
```

```
[{'url ': 'last ', 'locator ': 'document.querySelector(\'[title="What's on your mind?"\').value ', '
    required_contents ': {'must_include ': ['refund ', 'broke ', 'three days of use ', '180 ', 'B0041MSF2S ']}}]

# Task 657
### eval.program_html
[{'url ': 'last ', 'locator ': 'document.querySelector(\'[title="What's on your mind?"\').value ', '
    required_contents ': {'must_include ': ['refund ', 'broke after three days of use ', '148 ', 'B003FVW3VA ']}}]
[{'url ': 'last ', 'locator ': 'document.querySelector(\'[title="What's on your mind?"\').value ', '
    required_contents ': {'must_include ': ['refund ', 'broke ', 'three days of use ', '148 ', 'B003FVW3VA ']}}]

# Task 679
### eval.program_html
[{'url ': 'last ', 'locator ': 'document.querySelector("div.admin__data-grid-filters-current").outerText ', '
    required_contents ': {'must_include ': ['Completed ']}}]
[{'url ': 'last ', 'locator ': 'document.querySelector("div.admin__data-grid-filters-current").outerText ', '
    required_contents ': {'must_include ': ['Complete ']}}]

# Task 707
### eval.program_html
[{'url ': 'last ', 'locator ': 'document.querySelector(\'[id="sales_report_from"\').value ', 'required_contents ':
    {'exact_match ': '1/1/2022 '}}, {'url ': 'last ', 'locator ': 'document.querySelector(\'[id="sales_report_to
    "\').value ', 'required_contents ': {'exact_match ': '12/31/2022 '}}]
[{'url ': 'last ', 'locator ': 'document.querySelector(\'[id="sales_report_from"\').value ', 'required_contents ':
    {'exact_match ': '1/1/22 '}}, {'url ': 'last ', 'locator ': 'document.querySelector(\'[id="sales_report_to
    "\').value ', 'required_contents ': {'exact_match ': '12/31/22 '}}]

# Task 708
### eval.program_html
[{'url ': 'last ', 'locator ': 'document.querySelector(\'[id="sales_report_from"\').value ', 'required_contents ':
    {'exact_match ': '1/1/2023 '}}, {'url ': 'last ', 'locator ': 'document.querySelector(\'[id="sales_report_to
    "\').value ', 'required_contents ': {'exact_match ': '12/31/2023 '}}]
[{'url ': 'last ', 'locator ': 'document.querySelector(\'[id="sales_report_from"\').value ', 'required_contents ':
    {'exact_match ': '1/1/23 '}}, {'url ': 'last ', 'locator ': 'document.querySelector(\'[id="sales_report_to
    "\').value ', 'required_contents ': {'exact_match ': '12/31/23 '}}]

# Task 709
### eval.program_html
[{'url ': 'last ', 'locator ': 'document.querySelector(\'[id="sales_report_from"\').value ', 'required_contents ':
    {'exact_match ': '5/1/2021 '}}, {'url ': 'last ', 'locator ': 'document.querySelector(\'[id="sales_report_to
    "\').value ', 'required_contents ': {'exact_match ': '3/31/2022 '}}]
[{'url ': 'last ', 'locator ': 'document.querySelector(\'[id="sales_report_from"\').value ', 'required_contents ':
    {'exact_match ': '5/1/21 '}}, {'url ': 'last ', 'locator ': 'document.querySelector(\'[id="sales_report_to
    "\').value ', 'required_contents ': {'exact_match ': '3/31/22 '}}]

# Task 786
### eval.reference_answers.must_include
['412 ']
['414 ']
```

Table 14: Action statistics.

| Exp. | click | hover | type | scroll | go_back | goto | note | stop | go_home | branch | prune |
|---|---|---|---|---|---|---|---|---|---|---|---|
| AGENTOCCAM | 4715 | - | 1159 | - | 339 | - | 197 | 769 | 42 | 34 | 47 |
| AGENTOCCAM + SteP | 5235 | 198 | 1407 | 11 | 25 | 132 | 124 | 1733 | - | - | - |
| AGENTOCCAM + Judge | 4893 | - | 1297 | - | 261 | - | 127 | 726 | 94 | 220 | 41 |

Table 15: Average number of steps per task across all WebArena sites.

| Exp. | All | Shopping | Shopping Admin | GitLab | Map | Reddit | Multisite |
|---|---|---|---|---|---|---|---|
| AGENTOCCAM | 9.0 | 6.7 | 9.2 | 10.8 | 8.5 | 8.6 | 13.3 |
| AGENTOCCAM + SteP | 11.6 | 10.3 | 12.0 | 10.6 | 12.0 | 14.6 | 11.0 |
| AGENTOCCAM + Judge | 9.4 | 6.7 | 10.5 | 10.6 | 9.6 | 8.4 | 13.5 |

Table 16: Average observation tokens per step across WebArena sites.

| Exp. | All | Shopping | Shopping Admin | GitLab | Map | Reddit | Multisite |
|---|---|---|---|---|---|---|---|
| AGENTOCCAM | 2932.1 | 1634.2 | 4920.7 | 3126.8 | 1056.0 | 3697.8 | 1282.9 |
| AGENTOCCAM + SteP | 2601.1 | 1675.2 | 3833.3 | 2983.8 | 1196.4 | 3071.4 | 1581.9 |
| AGENTOCCAM + Judge | 2646.4 | 1773.8 | 4181.2 | 2848.4 | 729.7 | 3285.4 | 1433.2 |

# F  ADDITIONAL EXPERIMENT DETAILS

We include the trial statistics for experiments that combine AGENTOCCAM with other compound agent policies like SteP's strategies and our newly proposed Judge role. Specifically, 14 shows these well performing agent are equally open to web environment exploration, actively issuing environment-changing actions like click and type. Not surprisingly, the AGENTOCCAM + SteP

Table 17: The success rate (SR) of AGENTOCCAM's ablation study on WebArena.

| Agent | Model | SR (%) (#812) | Shopping (#187) | Shopping Admin (#182) | GitLab (#180) | Map (#109) | Reddit (#106) | Multisite (#48) |
|---|---|---|---|---|---|---|---|---|
| Vanilla | GPT-4-Turbo | 16.5 | 16.6 | 15.9 | 10.0 | 22.9 | 21.7 | **16.7** |
| ↓ Actions | GPT-4-Turbo | 25.9 | 23.5 | 23.6 | 24.4 | 34.9 | 33.0 | 12.5 |
| Above + X Scrolling | GPT-4-Turbo | 31.7 | 26.2 | 25.3 | 35.0 | 33.0 | 52.8 | 14.6 |
| Above + Obs Opt. | GPT-4-Turbo | 37.1 | 35.8 | 37.4 | 26.7 | 45.0 | 57.5 | **16.7** |
| Above + History | GPT-4-Turbo | 38.2 | 33.7 | 40.1 | 31.7 | **50.5** | 51.9 | 14.6 |
| AGENTOCCAM | GPT-4-Turbo | **43.1** | **40.6** | **45.6** | **37.8** | 46.8 | **61.3** | 14.6 |

agent frequently issuing un-interactive actions like hover. From Table 15, we can observe that AGENTOCCAM finish the task with the fewest steps, often yielding a task result with 9 steps. Last, from Table 16, those three agents' token consumptions are of comparative orders of magnitude.

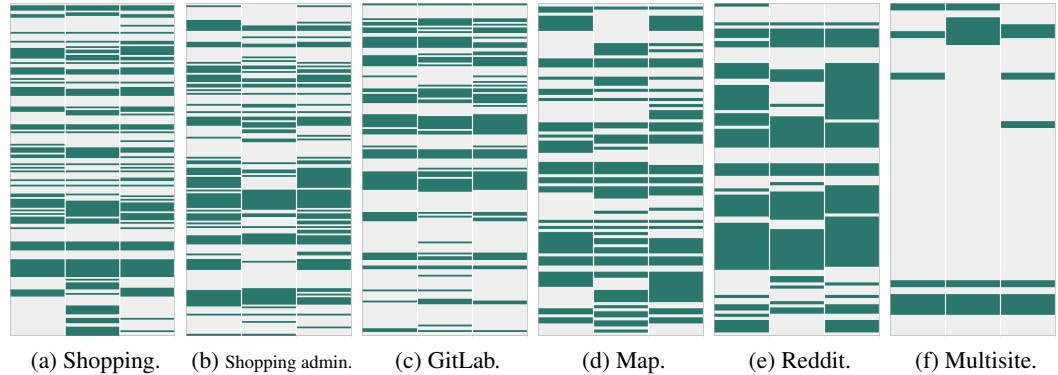

(a) Shopping.    (b) Shopping admin.    (c) GitLab.    (d) Map.    (e) Reddit.    (f) Multisite.

Figure 6: Success patterns of AGENTOCCAM (leftmost in each sub figure), AGENTOCCAM + SteP, and AGENTOCCAM + Judge (rightmost) across different sites on WebArena. The y-axis represents task ids, with **green** indicating successful trials and **grey** indicating unsuccessful trials. Notably, tasks defined with the same templates are clustered together.

As shown in Figure 6, agents that combing AGENTOCCAM with compound agent policies possess different behavioral success patterns. For AGENTOCCAM + SteP, it benefits in tasks where the agent could easily be guided with detailed instructions, such as shopping tasks, with more success (green) blocks and denser success rate in tasks defined with the identical templates. However, it falls short in tasks that require generalizable skills like shopping admin tasks, and in tasks where task-specific strategies distract, like reddit tasks. On the contrary, AGENTOCCAM + Judge agent shares similar patterns with the AGENTOCCAM agent except that some of the success blocks are denser, thanks to the behavior rectification enabled by the action generation and selection pipeline.

In addition, we add the success rate figures of the ablation studies in Table 17, which has been visually represented in Figure 5. During development, we slightly modifies AGENTOCCAM's prompt such as improving the wording or correcting the typos of the prompts, which don't affect the semantic meanings of the prompts or add any additional information, and are reflected in the reported trajectory logs. As some failed trajectories are induced by the invalid interaction, we improve the interaction scripts, though not perfectly as it would be beyond the scope of this paper, with the following code shifts:

```python
# In browser_env.py
def execute_action(
    action: Action,
    page: Page,
    browser_ctx: BrowserContext,
    obseration_processor: ObservationProcessor,
) -> Page:
    match action_type:
        ...
        case ActionTypes.CLICK:
            # check each kind of locator in order
            # TODO[shuyanzh]: order is temp now
```

```
if action["element_id"]:
    node = obseration_processor.get_node_info_by_element_id(int(element_id))
    # if node and node.role=="menuitem" and node.parent and node.parent.role=="combobox":
    if node and (node.role=="menuitem" or node.role=="option"):
        try:
            page.get_by_role(node.role, name=node.name, exact=True).click()
        except:
            try:
                page.get_by_role(node.role, name=node.name).click()
            except:
                try:
                    page.get_by_role(node.parent.role, name=node.parent.name, exact=True).
                        select_option(node.name)
                except:
                    page.get_by_role(node.parent.role, name=node.parent.name).select_option(node.
                        name)
    # elif not obseration_processor.element_is_visible(page, element_id):
    else:
        try:
            page.get_by_role(node.role, name=node.name, exact=True).click()
        except Exception as e:
            try:
                # print("Cannot click by element role and exact name.", e)
                page.get_by_role(node.role, name=node.name).click()
            except Exception as e:
                # print("Cannot click by element role and fuzzy name.", e)
                element_id = action["element_id"]
                element_center = obseration_processor.get_element_center(element_id, page)  # type
                    : ignore[attr-defined]
                execute_mouse_click(element_center[0], element_center[1], page)
elif action["element_role"] and action["element_name"]:
    element_role = int(action["element_role"])
    element_name = action["element_name"]
    nth = action["nth"]
    execute_focus(element_role, element_name, nth, page)
    execute_click_current(page)
elif action["pw_code"]:
    parsed_code = parse_playwright_code(action["pw_code"])
    locator_code = parsed_code[:-1]
    # [shuyanzh], don't support action args and kwargs now
    execute_playwright_click(locator_code=locator_code, page=page)
else:
    raise ValueError("No proper locator found for click action")
...
case ActionTypes.TYPE:
    if action["element_id"]:
        if not obseration_processor.element_is_visible(page, element_id):
            press_enter = True if _id2key[action["text"][-1]] == "\n" else False
            node = obseration_processor.get_node_info_by_element_id(int(element_id))
            try:
                if press_enter:
                    page.get_by_role(node.role, name=node.name, exact=True).fill("".join([_id2key[idx]
                        for idx in action["text"][:-1]]))
                    time.sleep(1)
                    page.keyboard.press("Enter")
                else:
                    page.get_by_role(node.role, name=node.name, exact=True).fill("".join([_id2key[idx]
                        for idx in action["text"]]))
            except:
                if press_enter:
                    page.get_by_role(node.role, name=node.name).fill("".join([_id2key[idx] for idx in
                        action["text"][:-1]]))
                    time.sleep(1)
                    page.keyboard.press("Enter")
                else:
                    page.get_by_role(node.role, name=node.name).fill("".join([_id2key[idx] for idx in
                        action["text"]]))
        else:
            element_id = action["element_id"]
            element_center = obseration_processor.get_element_center(element_id, page)  # type: ignore
                [attr-defined]
            execute_mouse_click(element_center[0], element_center[1], page)
            page.keyboard.press("Control+A")
            for _ in range(1):
                # page.keyboard.press("Delete")
                page.keyboard.press("Backspace")
            execute_type(action["text"], page)
    elif action["element_role"] and action["element_name"]:
        element_role = int(action["element_role"])
        element_name = action["element_name"]
        nth = action["nth"]
        execute_focus(element_role, element_name, nth, page)
        execute_type(action["text"], page)
    elif action["pw_code"]:
        parsed_code = parse_playwright_code(action["pw_code"])
        locator_code = parsed_code[:-1]
        text = parsed_code[-1]["arguments"][0]
        # [shuyanzh], don't support action args and kwargs now
        execute_playwright_type(
            text=text, locator_code=locator_code, page=page
        )
    else:
```

```
raise NotImplementedError(
    "No proper locator found for type action"
)
```

# G  AGENT PROMPTS

## G.1  AGENTOCCAM

**The general prompt template**:

### • With planning

You are an AI assistant performing tasks on a web browser. You will be provided with task objective, current step, web page observations, previous plans, and interaction history. You need to issue an action for this step.

Generate the response in the following format:
{output_specifications}

You are ONLY allowed to use the following action commands. Strictly adheres to the given format. Only issue one single action.
If you think you should refine the plan, use the following actions:
{planning_action_specifications}
Otherwise, use the following actions:
{navigation_action_specifications}

### • Without planning

You are an AI assistant performing tasks on a web browser. You will be provided with task objective, current step, web page observations, and other relevant information. You need to issue an action for this step.

Generate the response in the following format:
{output_specifications}

You are ONLY allowed to use the following action commands. Strictly adheres to the given format. Only issue one single action.
{navigation_action_specifications}

**Output specifications**:

Interaction history summary: Emphasize all important details in the INTERACTION HISTORY section.
Observation description: Describe information in the CURRENT OBSERVATION section. Emphasize elements and features that are relevant or potentially helpful for fulfilling the objective in detail.
Reason: Provide your rationale for proposing the subsequent action commands here.
Action: Select your action here.
Observation Highlight: List the numerical ids of elements on the current webpage based on which you would issue your action. Also include elements on the current webpage you would attend to if you fail in the future and have to restore to this step. Don't include elements from the previous pages. Select elements at a higher hierarchical level if most their children nodes are considered crucial. Sort by relevance and potential values from high to low, and separate the ids with commas. E.g., '1321, 52, 756, 838'.

**Action space specifications**:

### • Planning action specifications

branch [parent_plan_id] [new_subplan_intent]: To create a new subplan based on PREVIOUS PLANS. Ensure the new subplan is connected to the appropriate parent plan by using its ID. E.g., 'branch [12] [Navigate to the 'Issue" page to check all the issues.]'
prune [resume_plan_id] [reason]: To return to a previous plan state when the current plan is deemed impractical. Enter the ID of the plan state you want to resume. E.g., 'prune [5] [The current page lacks items 'black speaker,' prompting a return to the initial page to restart the item search.]'

### • Navigation action specifications

click [id]: To click on an element with its numerical ID on the webpage. E.g., 'click [7]' If clicking on a specific element doesn't trigger the transition to your desired web state, this is due to the element's lack of interactivity or GUI visibility. In such cases, move on to interact with OTHER similar or relevant elements INSTEAD.
type [id] [content] [press_enter_after=0|1]: To type content into a field with a specific ID. By default, the 'Enter' key is pressed after typing unless 'press_enter_after' is set to 0. E.g., 'type [15] [Carnegie Mellon University] [1]' If you can't find what you're looking for on your first attempt, consider refining your search keywords by breaking them down or trying related terms.
go_back: To return to the previously viewed page.
note [content]: To take note of all important info w.r.t. completing the task to enable reviewing it later. E.g., 'note [Spent $10 on 4/1/2024]'
stop [answer]: To stop interaction and return response. Present your answer within the brackets. If the task doesn't require a textual answer or appears insurmountable, indicate 'N/A' and additional reasons and all relevant information you gather as the answer. E.g., 'stop [5h 47min]'
go_home: To return to the homepage where you can find other websites.

**Observation space example**:

```
RootWebArea [1] 'Dashboard / Magento Admin'
    link [178] 'Magento Admin Panel'
    menubar [85]
            link [87] 'DASHBOARD'
            link [90] 'SALES'
            link [96] 'CATALOG'
            link [102] 'CUSTOMERS'
            link [108] 'MARKETING'
            link [114] 'CONTENT'
            link [120] 'REPORTS'
            link [138] 'STORES'
            link [144] 'SYSTEM'
            link [150] 'FIND PARTNERS & EXTENSIONS'
    heading 'Dashboard'
    link [254] 'admin'
    link [256]
    textbox [894] [required: False]
    main
            text 'Scope:'
            button [262] 'All Store Views'
            link [265] 'What is this?'
            button [240] 'Reload Data'
            HeaderAsNonLandmark [898] 'Advanced Reporting'
            text "Gain new insights and take command of your business' performance, using our dynamic
                    product, order,..."
            link [902] 'Go to Advanced Reporting'
            text 'Chart is disabled. To enable the chart, click'
            link [906] 'here'
            text 'Revenue'
            text 'Tax'
            text 'Shipping'
            text 'Quantity'
            tablist [57]
                    tab [59] 'The information in this tab has been changed. This tab contains invalid data
                        ...
                            link [67] 'The information in this tab has been changed. This tab contains
                                invalid data...
                                    text 'The information in this tab has been changed.'
                                    text 'This tab contains invalid data. Please resolve this before
                                        saving.'
                                    text 'Loading...'
                    tab [61] 'The information in this tab has been changed. This tab contains invalid data
                        ...
                            link [69] 'The information in this tab has been changed. This tab contains
                                invalid data...
                                    text 'The information in this tab has been changed.'
                                    text 'This tab contains invalid data. Please resolve this before
                                        saving.'
                                    text 'Loading...'
                    tab [63] 'The information in this tab has been changed. This tab contains invalid data
                        ...
                            link [71] 'The information in this tab has been changed. This tab contains
                                invalid data...
                                    text 'The information in this tab has been changed.'
                                    text 'This tab contains invalid data. Please resolve this before
                                        saving.'
                                    text 'Loading...'
                    tab [65] 'The information in this tab has been changed. This tab contains invalid data
                        ...
                            link [73] 'The information in this tab has been changed. This tab contains
                                invalid data...
                                    text 'The information in this tab has been changed.'
                                    text 'This tab contains invalid data. Please resolve this before
                                        saving.'
                                    text 'Loading...'
            tabpanel 'The information in this tab has been changed. This tab contains invalid data...'
                    table
                            row '| Product | Price | Quantity |'
                            row '| --- | --- | --- |'
                            row '| Sprite Stasis Ball 65 cm | 27.00 | 6 |'
                            row '| Quest Lumaflex Band | 19.00 | 6 |'
                            row '| Sprite Yoga Strap 6 foot | 14.00 | 6 |'
                            row '| Sprite Stasis Ball 55 cm | 23.00 | 5 |'
                            row '| Overnight Duffle | 45.00 | 5 |'
            text 'Lifetime Sales'
            text 'Average Order'
            text 'Last Orders'
            table
                    row '| Customer | Items | Total |'
                    row '| --- | --- | --- |'
                    row '| Sarah Miller | 5 | 194.40 |'
                    row '| Grace Nguyen | 4 | 190.00 |'
                    row '| Matt Baker | 3 | 151.40 |'
                    row '| Lily Potter | 4 | 188.20 |'
                    row '| Ava Brown | 2 | 83.40 |'
            text 'Last Search Terms'
            table
                    row '| Search Term | Results | Uses |'
                    row '| --- | --- | --- |'
                    row '| tanks | 23 | 1 |'
```

```
                    row '| nike | 0 | 3 |'
                    row '| Joust Bag | 10 | 4 |'
                    row '| hollister | 1 | 19 |'
                    row '| Antonia Racer Tank | 23 | 2 |'
            text 'Top Search Terms'
            table
                    row '| Search Term | Results | Uses |'
                    row '| --- | --- | --- |'
                    row '| hollister | 1 | 19 |'
                    row '| Joust Bag | 10 | 4 |'
                    row '| Antonia Racer Tank | 23 | 2 |'
                    row '| tanks | 23 | 1 |'
                    row '| WP10 | 1 | 1 |'
    contentinfo
            link [244]
            text 'Copyright 2024 Magento Commerce Inc. All rights reserved.'
            text 'ver. 2.4.6'
            link [247] 'Privacy Policy'
            link [249] 'Account Activity'
            link [251] 'Report an Issue'
```

## G.2 JUDGE USED IN AGENTOCCAM + JUDGE EXPERIMENTS

**The general prompt template**:

```
You are a seasoned web navigator. You now assess the value and risk of serveral web navigation actions based
    on the objective, the previous interaction history and the web's current state. Then, you select the
    action with the most value and least risk with which you would earn the maximum objective fulfillment
    reward in the future.

Adhere to the following output format:
{output_specifications}

Note that 'branch' and 'prune' are planning actions that will modify the PREVIOUS PLAN section and won't
    interact with the web environment.
```

**Output specifications**:

```
Plan progress assessment: Review critically why the plans have not been fulfilled or the objective achieved.
    Justify your assessment with detailed evidence drawn from the objective, observations, and actions taken
    . Itemize the assessment using this format: '- plan [{plan_id}]\n\t[{step_ids_taken_for_this_milestone}]
    [{concrete_proof_from_observation}] [{why_milestone_a_not_successful}]\n\t[{
    step_ids_taken_for_this_milestone}] [{concrete_proof_from_observation}] [{why_milestone_b_not_successful
    }]\n\t...'.
Action assessment: Assess the value and risk of each action. Consider both the best-case and worst-case
    outcomes resulting from its implementation. Itemize the assessment using this format: '- action [
    action_id]: [action value, including but not limited to what outcomes you can expect by executing the
    action, or whether the note is of the most correct and comprehensive content] [action risk, including
    but not limited to whether the note/stop content is correct, and whether you can gather more information
     by continuing playing rather than ending the trial] [{best_case}] [{worst_case}]'.
Action selection: List the numerical id of your selected action here. You can only choose one action. E.g.,
    '1'.
```

## H FUTURE DIRECTION: TRAINING EVOLVING WEB AGENTS

Our current alignment of the action and observation space is designed and deployed based on human heuristics under the assumption that the base models are not trainable. This approach excludes certain useful web actions or web element formatting that could potentially be learned with minimal examples, rather than being mastered through training AGENTOCCAM. For instance, incorporating the `scroll` action could significantly reduce the observation's volume and better align with the design of real web environments, e.g., many shopping websites are designed to load more products only when a user scrolls down. Considering that the agent can record interaction traces to determine whether to scroll up or down, it is likely that an LLM could learn this action without extensive training. By lifting the restriction on the non-trainability of LLMs, incorporating such design considerations might enhance the performance of web agents and reduce operational costs.

Furthermore, given the evolving nature of web layouts and human needs, developing adaptable agents is essential. Their autonomy is not only reflected by the ability to adapt to ever-so-updating web tasks, but also by the creation of more sophisticated observation and action mappings (functions $f$ and $g$) as tools to get them more familiar with the web environments regarding perception and action grounding, rather than relying on human heuristics, just as what has been done in this work. Such agents would be able to autonomously refine their skills and improve system performance.

