# OpenReview forum: "AgentOccam: A Simple Yet Strong Baseline for LLM-Based Web Agents"
_ICLR.cc/2025/Conference — ICLR 2025 Poster_

### Official Review · Reviewer_Tca3 · 2024-10-27

**Soundness:** 2
**Presentation:** 2
**Contribution:** 2
**Rating:** 6
**Confidence:** 4

**Summary:**

This paper proposes a new pathway to improve the performance of LLM-based web agents. Compared with previous works on web agents, it mainly focuses on optimizing the action space and observation space, rather than designing reasoning/planning/memory modules for more complex inferences. The performance on WebArena shows the effectiveness of AgentOccam on web tasks.

**Strengths:**

Pros:
- The perspective to improve the capability of web agents is great, which it dives into the action space and observation space. It is a common pathway that may take effect in not only web tasks but also other tasks with complex actions and observations.
- I think the action optimization is inspired, especially for abstracting and generating new actions. From my perspective, it is similar to the tool learning problem and can be effective.
- The improvement in performances seems great, where it demonstrates a huge improvement on WebArena.

**Weaknesses:**

Cons:
- The author claims to solve the generalizability across all real-world applications in the abstract. However, the method section seems to just illustrate how to deal with the specific web environment. There are too many 'specifically' and 'in particular' in sections 4.1 and 4.2. Does it mean I need to design different merges manually, where the method is just providing a guide or strategy to conduct the manual process? I think an ideal optimization is to conduct the optimization automatically, even by prompting LLMs. You may provide concrete examples of how your approach could be applied to other domains, or discuss any limitations in generalizability.
- I think a general method to automatically conduct the alignment is necessary, especially for the action space mentioned above. But this work seems to fail to do so.
- I'm confused about the part 'Replaying Observation History Selectively', could you please demonstrate with more details? I think this is an information retrieval process, but I'm aware of how to get relevant information. You may provide a step-by-step example of how the selective replay process works on a sample task. This would help clarify the information retrieval aspect that I'm unsure about. Additionally, you might show more details on how the relevance of information is determined and how this process differs from or improves upon standard information retrieval techniques.

**Questions:**

Please see the cons. Please point out if I have any misunderstandings, and I'm willing to improve my rating if the author could greatly address my concerns.

---

> ### Author Response · Authors · 2024-11-22
>
> We greatly appreciate your feedback. It's heartening to learn that you recognize our work's contributions of  proposing the widely applicable idea of action and observation space optimization of large language model (LLM) agents, different from previous work's focus on designing other modules. Your positive remarks about the inspiration from our action optimization, akin to effective tool learning, are highly valued. **We have updated the manuscript according to your feedback, and have added code for all experiments and new experiment trajectories to the supplementary materials.** We organize our responses to your questions in the order they are presented and hope they address any concerns you may have.
>
> ## AgentOccam's Domain Generalizabiliy
> We are very delighted that you perceive the inherent generalizability of our proposed action and observation space alignment with LLM training to other agent tasks. However, our agent policy introduced in the paper is specifically applied to executing general-purpose web tasks, because in those tasks the agent's action/observation space deviates significantly from the LLMs' training data, and thus the task representation alignment could yield the most remarkable performance gain. Our methods proposed in Sections 4.1 and 4.2 are completely automated with deterministic rules, and we upload AgentOccam's code as the rebuttal supplementary materials in case you would like to play it in any web environment.
>
> Additionally, we help conduct experiments on the general-purpose web task benchmark WebVoyager and compare AgentOccam's performance with the benchmarks' SOTA Agent-E. We kindly refer you to the general response section for the details. In short, AgentOccam performs better than Agent-E on the WebVoyager's definite-answer-subset. Specifically, both agents have their specificities. We find that Agent-E has strong performance on websites like Wolfram Alpha with more delicate plans issued by its "planner," and AgentOccam outperforms it on websites like Google Search and Hugging Face which need more accurate task interpretation and information retrieval. The impressive results across 10+ websites and many types of tasks can show AgentOccam's generalizability in the web domain.
>
> Regarding how the principles of action and observation space alignment might help in other agent tasks, we provide a concrete example here for illustration.
>
> The ALFWorld benchmark (https://arxiv.org/abs/2010.03768) includes text-based virtual household tasks. It requires the agent to reason over the text description of the house environment to complete the tasks, with an example as follows:
>
> ```bash
> You are in the middle of a room. Looking quickly around you, you see a drawer 2, a shelf 5, a drawer 1, a shelf 4, a sidetable 1, a drawer 5, a shelf 6, a shelf 1, a shelf 9, a cabinet 2, a sofa 1, a cabinet 1, a shelf 3, a cabinet 3, a drawer 3, a shelf 11, a shelf 2, a shelf 10, a dresser 1, a shelf 12, a garbagecan 1, a armchair 1, a cabinet 4, a shelf 7, a shelf 8, a safe 1, and a drawer 4.
> Your task is to: put some vase in safe.
> ```
>
> The original observation is represented in sequential texts, enumerating the items around. This implicitly assigns an order to the objects, defying our intuitive impression on the biologically plausible environment perception, which should be that all the sensings happen and affect our decision making simultaneously. This can be corrected by representing the observation in the form of structured code:
>
> ```bash
> class Env():
> 	def __init__():
> 		self.drawer_idxs = {2, 1, 5, 3, 4}
> 		self.shelf_idxs = {5, 4, 6, 1, 9, 3, 11, 2, 10, 12, 7, 8}
> 		…
> def househould_task():
> 	# define the function of putting the vase in the safe here.
> 	pass
> ```
>
> Afterwards, we can prompt the LLMs to generate household procedures following the `househould_task()` functions defined.
>
> As we didn't conduct experiments on agent benchmarks other than web interaction, we will avoid overclaiming the generalizability of AgentOccam to those tasks in the revised paper.

---

> ### Author Response · Authors · 2024-11-22
>
> ## Alignment automation
> We appreciate your insights regarding the automation of observation and action space alignment. We would like to clarify that our observation space alignment is realized with hard-coded rules that can be automatically applied to any web environment, which we believe is a wide enough range of applications. Our action space alignment follows the exploration-first and less-embodiment heuristics that improves agent performance significantly and thus will potentially inspire future alignment automation research.
>
> To be more specific, when we are aligning the observation space, our input is the web page's DOM, which enables us to flexible merge or reshape elements based on their role (e.g., button, static text, table etc.), name (e.g., the text description of a button's functionality like submit or cancel, the content in a table cell, etc.), and other properties (e.g., focused, selected, etc.). All the procedures detailed in 4.2 "simplifying web page observations" can find a corresponding code snippet in the `AgentOccam/obs_opt.py` file that automates the observation processing. For example, merging the elements corresponds to the `action_merge_statictext_to_parent`, `action_merge_menuitem_and_option` etc. functions, and reformatting tables corresponds to the `action_reformat_table` function. Moreover, for the action space, our manually-designed alignment strategies are grounded in action statistics in Table 3 and the empirical action's exploration returns, which means we decide to retain or remove actions based on whether the action type encourages the agent to interact with the environment that leads to success, and whether it poses a challenge for the LLMs to interpret and use. These heuristics have been proven practical with experiment results, and we expect that they can guide action alignment automation principle designs in the future work.
>
> We thank you for pointing out plenty of promising future directions (especially the web agent's evolving tool learning) in our spirit, and we want to highlight that AgentOccam can serve as the basic step one for many web agents of this kind, including using LLM to process and generate observations and actions.

---

> ### Author Response · Authors · 2024-11-22
>
> ## "Replaying Observation History Selectively" Clarifications
> To prevent confusion, we separate the following explanations for the independent techniques of "selectively replaying web elements on a page" and "selectively replaying entire pages." We have updated the paper in Section 4.2 and in the appendix with these clarifications to make it a clearer version.
>
> ### Selectively Replaying Web Elements on a Page with Pivotal Nodes (Figure 4 Step 2)
> We will use how AgentOccam performed a simple customized task as an example for showing how we get the pivotal nodes and how those nodes help in selecting web elements for replaying.
>
> #### Intuition and Background
> We could view selectively replaying web elements on a page as a focused memory mechanism, where only the information that's relevant to the task would be recorded and replayed as the agent history traces. As all the web pages could be framed into a (DOM) tree structure, any web elements are nodes in this representation. We define pivotal nodes as the web elements, be it interactable or not, that are potentially useful for completing the task. Those pivotal nodes might present information (e.g., customer reviews in a review summary task) or the agent could interact with them to navigate to crucial states (e.g., the search box and the search button in a search task). We obtain these nodes at each step by prompting the agent to generate the page's highlights based on which they issue the action, or any web elements they will attend to if they fail at future steps and restore to the current state. After identifying the pivotal nodes with the agent's efforts, our code supports automatically parsing the web page's DOM tree and retaining nodes that are associated with the pivotal nodes (i.e., pivotal nodes' ancestor, sibling, and descendant nodes, as shown in Figure 4), to get a more succinct but less noisy version of the observation. This version would be used for constructing the agent's focused working history.
>
> #### Prompt, Pivotal Node Section
> ```bash
> Generate the response in the following format:
> …
> OBSERVATION HIGHLIGHT:
> List the numerical ids of elements on the current webpage based on which you would issue your action. Also include elements on the current webpage you would attend to if you fail in the future and have to restore to this step. Don't include elements from the previous pages. Select elements at a higher hierarchical level if most their children nodes are considered crucial. Sort by relevance and potential values from high to low, and separate the ids with commas. E.g., `1321, 52, 756, 838`.
> ```
>
> #### Example Task Objective
> What is the email address of the Dean of the School of Engineering at Stanford University?

---

> ### Author Response · Authors · 2024-11-22
>
> ### Selectively Replaying Web Elements on a Page with Pivotal Nodes (Figure 4 Step 2) (Contd.)
> #### AgentOccam's Workflow Regarding the Pivotal Nodes
> 0. The task started at the Google.com, with the observation to be:
> ```
> RootWebArea 'Google'
> 	link [29] 'About'
> 	link [30] 'Store'
> 	link [277] 'Gmail'
> 	link [275] 'Search for Images'
> 	button [282] 'Google apps'
> 	link [152] 'Sign in'
> 	IframePresentational [153]
> 	search [6]
> 		combobox [12] 'Search' [required: False]
> 		button [294] 'Search by voice'
> 		button [295] 'Search by image'
> 		button [272] 'Google Search'
> 		button [273] "I'm Feeling Lucky"
> 	contentinfo
> 		link [83] 'Advertising'
> 		link [84] 'Business'
> 		link [85] 'How Search works'
> 		link [89] 'Our third decade of climate action: join us'
> 		link [93] 'Privacy'
> 		link [94] 'Terms'
> 		button [100] 'Settings'
> 			generic [102] 'Settings'
> ```
> 1. The agent typed the keyword into the search box and identified element 12 (combobox [12] 'Search' [required: False]) and 272 (button [272] 'Google Search') as the pivotal nodes. The web transit to a Google search page with the searched entries listed. Now, the agent would be prompted with task history and web observation for issuing the next action, where the task history was constructed based on the pivotal nodes selected by the agent previously, and any pivotal-node-associated nodes. We take the automatically-generated interaction history clip from the prompt at this step:
> ```
> <step_0_interaction>
> OBSERVATION:
> RootWebArea 'Google'
> 	search
> 		combobox 'Search' [required: False]
> 		button 'Search by voice'
> 		button 'Search by image'
> 		button 'Google Search'
> 		button "I'm Feeling Lucky"
> REASON FOR ACTION:
> To progress towards finding the email address of the Dean of the School of Engineering at Stanford University, the first logical step is to search for relevant information. Using the search input box to enter a query such as "Dean of the School of Engineering Stanford University email" and then submitting it using the 'Google Search' button is a direct approach to gather this information.
> ACTION:
> type [12] [Dean of the School of Engineering Stanford University email] [1]
> </step_0_interaction>
> ```
> 2. We can observe that the "pivotal node" technique helps keep a focused memory attention by only retaining the context of crucial information and omitting irrelevant noise. Based on the interaction history summary and current observation, the agent executed the following steps, with the history section constantly updated with the information from the ongoing steps till trajectory ended.

---

> ### Author Response · Authors · 2024-11-22
>
> ## "Replaying Observation History Selectively" Clarifications (Contd.)
> ### Selectively Replaying Entire Pages with the Planning Tree (Figure 4 Step 3)
> We will use a planning tree generated by our agent during the development stage as an example for explaining how the planning tree takes shape and helps in selectively replaying past pages.
>
> #### Intuition and Background
> If we make an analogy of the single-thread web task completion to code execution, the agent's issuing sub plans mimic nested function calling, the active plan is like the uppermost function in the execution stack, and pruning the planning tree plays as popping failed functions off the stack. There constantly exists a planning tree and the planning tree operation instructions in the agent's prompt. We enable the agent to organize the planning tree with `branch` and `prune` commands, where the `branch` action creates new subplans, and the `prune` action restores the task progress to a previous state. It's noteworthy that those planning commands are in the same position as navigation prompts for the agent.
>
> #### Prompt, Planning Tree Section
> ```bash
> If you think you should refine the plan, use the following actions:
> branch [parent_plan_id] [new_subplan_intent]: To create a new subplan based on PREVIOUS PLANS. Ensure the new subplan is connected to the appropriate parent plan by using its ID. E.g., `branch [12] [Navigate to the "Issue" page to check all the issues.]`
> prune [resume_plan_id] [reason]: To return to a previous plan state when the current plan is deemed impractical. Enter the ID of the plan state you want to resume. E.g., `prune [5] [The current page lacks items "black speaker," prompting a return to the initial page to restart the item search.]`
> Otherwise, use the following actions:
> {navigation_specifications}
> ```
>
> #### Example Task Objective
> (WebArena 174) Open my latest updated issue that has keyword "feature" in its title to check if it is closed.
>
> #### AgentOccam's Workflow Regarding the Planning Tree
> 0. At the beginning, we used the task objective as the root plan:
> ```
> [0] (Active Plan) Find the solution to "Open my latest updated issue that has keyword "feature" in its title to check if it is closed"
> ```
> 1. The agent added a subplan to planning node 0 by issuing `branch [0] [Navigate to the Issues page to search for the latest issue with the keyword "feature" in the title.]`. Now the planning tree changed into:
> ```
> [0] Find the solution to "Open my latest updated issue that has keyword "feature" in its title to check if it is closed"
> 	[1] (Active Plan) Navigate to the Issues page to search for the latest issue with the keyword "feature" in the title.
> ```
> 2. The agent navigated to the project's issue page.
> 3. The agent decomposed the plan by generating branch actions `branch [1] [Search for the latest issue with the keyword "feature" in the title and check if it is closed.]` and `branch [1] [Open the latest issue with the keyword "feature" in the title.]`. It performed navigation steps for each active plan before the next planning command was executed. In this example, after `branch [1] [Search for the latest issue with the keyword "feature" in the title and check if it is closed.]` was proposed, the active plan turned into "Search for the latest issue with the keyword "feature" in the title and check if it is closed." The agent then typed the keyword "feature" into the search box and sorted the issues by operating the sort icon before generating the next plan "Open the latest issue with the keyword "feature" in the title." In other words, all the navigation commands (e.g., search and sort) it issued were intended for the current active plan (e.g., search for the latest issue with the keyword "feature" in the title and check if it is closed). Just like a function call only needs to consider the function's scope, this allows the agent to only attend to the navigation actions dedicated to the active plan and the corresponding web observation as the playing history, which helps selectively replay past pages for the agent. Note that in this case, it assigned the two new sub plans to the same parent plan [1], which automatically shaped the planning tree's structure. Finally, the planning tree reformed into (the content enclosed in "[]" means comments, which is intended for illustration and didn't appear in the agent's prompt):
> ```
> [0] Find the solution to "Open my latest updated issue that has keyword "feature" in its title to check if it is closed"
> 	[1] Navigate to the Issues page to search for the latest issue with the keyword "feature" in the title. [Plan's action scope: navigate.]
> 		[2] Search for the latest issue with the keyword "feature" in the title and check if it is closed. [Plan's action scope: search and sort.]
> 		[3] (Active Plan) Open the latest issue with the keyword "feature" in the title.
> ```
> 4. The agent executed the following steps to complete the task.

---

> ### Author Response · Authors · 2024-11-22
>
> ## Miscs.
> If you have any additional questions or concerns, please let us know so we can resolve them before the discussion period concludes. Otherwise, it would be greatly appreciated if you could raise your score to show that the existing concerns have been addressed. Thank you!

---

> ### Author Response · Authors · 2024-11-25
>
> Dear Reviewer,
>
> Thank you once again for your time and effort in reviewing our paper.
>
> As the discussion period is nearing its end, we kindly remind you that there are two days remaining for any further comments or questions. We would be grateful for the chance to address any additional concerns before it concludes.
>
> Thank you again!
>
> Best regards,
>
> Authors

---

> > ### Comment · Reviewer_Tca3 · 2024-11-25
> >
> > Thanks for your detailed response. I have raised my score.

---

### Official Review · Reviewer_LC8w · 2024-11-03

**Soundness:** 3
**Presentation:** 3
**Contribution:** 2
**Rating:** 5
**Confidence:** 4

**Summary:**

The paper proposes a new method for improving LLM-based web agents. The contributions are mainly two folds:

1. Improve alignment between the action and observation space of the web agent and the data on which the LLM has been pre-trained. The authors propose heuristics to: a. Simplify the actions available to the agent and b. Better clean and format the web page observation input for the LLM.

2. New actions and improvement for planning. The authors introduce new `branch` and `prune` actions for altering the plan, as well as prune the web elements included in the historical context.

The authors conduct experiments with WebArena, a web agent benchmark that includes simulation environments across 5 websites.

- The proposed method obtains significant improvement over the base agent design from the original WebArena.
- The proposed method also outperforms the SOTA method on WebArena by 6 points. (Note that the base model is not the same)
- Through ablations, both the action/observation alignment and improvement in planning contribute to the overall bump in performance, with the most gain coming from the action/observation simplification.
- The proposed method can also be combined with another method though the improvement depends on specific agent design. In Step, the proposed method brings improvement compared with Step, though it does not bring further improvement to AgentOccam. When combined with an LLM judge, the authors observe further improvement over AgentOccam.

**Strengths:**

- The overall experiment results are positive, and new SOTA performance was obtained on the WebArena benchmark.
- The proposed design is simple yet effective, so it should be rather easy to be applied.
- The experiment on WebArena is extensive, with a comparison of multiple base models and ablations to show improvement of individual components.

**Weaknesses:**

- The main contribution, and also the main source of improvement, is the simplification of action and observation space. A natural question would be how well these simplifications could be generalized across websites and tasks. Although WebArena contains multiple tasks, it still only contains five websites. To further verify the effectiveness of the proposed method, I would suggest including experiments on other datasets, e.g., Mind2Web, which has more websites.

- Similarly, the alignment could potentially be LLM-dependent. The paper only conducted experiments with GPT-4; it would be better if it could be shown that the method could generalize to different LLMs, which potentially have seen different mixes and formats of training data.

- How to properly represent webpage observation and select the action space has been studied by a few works, both for web agents and web IE, in the past (1) (or, in general, all papers on web agents have touched on this topic). It is appreciated that the authors find a new representation that works very well, but the technical novelty is a bit limited, especially for conferences like ICLR.

1. Zhou Y, Sheng Y, Vo N, Edmonds N, Tata S. Simplified DOM Trees for Transferable Attribute Extraction from the Web.

**Questions:**

1. Are there experiments on more datasets to show the effectiveness of the proposed heuristic beyond WebArena?

2. Are there experiments with other LLMs.

3. If possible, it would be better to include results using the same base LLM. It is okay to report SOTA on the best LLM available. But for fair comparison and a better understanding of the contribution of the proposed method, it would be very helpful to include results using the same LLM as the baselines.

4. I recommend swapping Figure 5 and Table 10 for the ablation results. The numbers in the table is easier to consume.

---

> ### Author Response · Authors · 2024-11-22
>
> Thank you for your feedback. We are delighted to know that you recognize the contributions of our web agent's action and observation space alignment with the training of large language models (LLMs), and our introduction of `branch` and `prune` actions to enhance agent planning and task decomposition. We also appreciate that you conclude our experiments' key results and achievements. We would like to highlight that when we were designing and evaluating AgentOccam, the state-of-the-art (SOTA) method was SteP (https://arxiv.org/abs/2310.03720), and for a fair comparison with SOTA then, we reused their GPT model version (i.e., gpt-4-turbo). Additionally, it's noteworthy that all our experiments are conducted with gpt-4-turbo, whose fundamental performance is slightly worse than those versions the concurrent SOTA method use (i.e., gpt-4o for WebPilot (https://arxiv.org/abs/2408.15978)) (we refer to OpenAI's official model performance comparison: https://github.com/openai/simple-evals). **We have updated the manuscript according to your feedback, and have added code for all experiments and new experiment trajectories to the supplementary materials.** Your remaining questions are answered in sequence as raised.
>
> ## AgentOccam is Generalizable to Real-world Websites
> We kindly refer you to the general response section for the details. In short, AgentOccam performs better than Agent-E on the WebVoyager's definite-answer-subset. Specifically, both agents have their specificities. We find that Agent-E has strong performance on websites like Wolfram Alpha with more delicate plans issued by its "planner," and AgentOccam outperforms it on websites like Google Search and Hugging Face which need more accurate task interpretation and information retrieval. The impressive results across 10+ websites and many types of tasks can show AgentOccam's generalizability in the web domain.
>
> ## AgentOccam is Generalizable to different LLM Families
> In response to your question about AgentOccam’s LLM generalizability, we conduct the full set of ablation studies on a WebArena development subset with Gemini-1.5-flash (https://blog.google/technology/ai/google-gemini-ai/#sundar-note), a model trained with different data from the GPT model family. We also refer you to the general response section for the experiments' details. In conclusion, each alignment component proposed by AgentOccam proves beneficial to the agent system and can be generalized to different LLMs.

---

> ### Author Response · Authors · 2024-11-22
>
> ## AgentOccam's Technical Novelty
> We thank you for recommending web IE papers released before LLMs came out, which could complete the research background of web page information retrieval, and potentially enlighten future web-agent observation space optimization work. We will include those papers in related work in a future version of our paper..
>
> We would like to argue that in contrast to existing LLM-based agent designs, which emphasize compound agent policies as listed in Table 1, **our AgentOccam is the first to propose aligning task representation with LLM training to enhance agents' task understanding and reasoning, with extensive experiments and ablation studies on web tasks, where the agent's action/observation space deviates significantly from the LLMs' training data, and thus the task representation alignment could yield the most remarkable performance gain**. In short, our primary focus is on optimizing task representation for LLM-based web agents, which has the potential to inspire future work such as agent action selection automation and observation attention mechanisms. This marks the novelty of our work as the problem was overlooked by the LLM-based agent community, in 10+ existing web agent papers.
>
> **Previous web agent papers, which rely on additional modules, in context examples, task-specific strategies, offline data or online search, might need to adapt the agent's action and observation space to facilitate their policy designs, but their exploration is limited, or otherwise it cannot be explained that our AgentOccam outperforms them by a large margin without in context examples but with a consistent agent role and single LLM call at each step.**
>
> In addition, **compared with static web information extraction (IE) in the pre-LLM era, the agent's action and observation space refinement poses new challenges**, because i) Web IE works on extracting web attributes of interest, typically from the text field, the leaf node of the DOM tree, as implied in the paper you recommend (https://arxiv.org/abs/2101.02415), but the web agent is supposed to cater to all interactable/non interactable elements on a webpage, as it's required to not only extract relevant information but also dynamically interact with the web environment; and ii) LLM-based agent's observation space optimization covers more topics other than single-page refinement, such as agent planning and history organization, as those are included in prompt presented to the functioning LLMs. To address the above challenges, we propose to i) simplify web page elements by removing repetitive functionality- and layout-indicative tokens and reframing page content block in DOMs (not restricted to leaf nodes as in web IE), and replay observation history selectively with pivotal nodes and the planning tree, as detailed in Section 4. We also conduct extensive ablation studies to account for how each component of the action and observation space helps in agent task completion.

---

> ### Author Response · Authors · 2024-11-22
>
> ## Miscs.
> We appreciate your suggestion of swapping the visual illustration of the ablation study with the corresponding result table in the appendix. We will incorporate this in the final version. If you have any additional questions or concerns, please let us know so we can resolve them before the discussion period concludes. Otherwise, it would be greatly appreciated if you could raise your score to show that the existing concerns have been addressed. Thank you!

---

> ### Author Response · Authors · 2024-11-25
>
> Dear Reviewer,
>
> Thank you once again for your time and effort in reviewing our paper.
>
> As the discussion period is nearing its end, we kindly remind you that there are two days remaining for any further comments or questions. We would be grateful for the chance to address any additional concerns before it concludes.
>
> Thank you again!
>
> Best regards,
>
> Authors

---

> > ### Comment · Reviewer_LC8w · 2024-11-25
> > **Response to rebuttal**
> >
> > Thanks for providing the additional experiments and detailed rebuttal, I really appreciated it; I think it definitely made the submission stronger.
> >
> > However, my main concern is still the technical contribution and the generalization aspect. It is nice to have extra experiments with performance on more websites, but it also shows that out of 13 websites, the proposed method is better in 6, tied in 3, and worse in 4. While it still shows an edge, it does not offer a conclusive answer as to whether the method would face challenges in generalization and whether there are common characteristics for the websites where the proposed method performs worse.
> >
> > Also, the method is a bit tied to the current web agent design and base model capability. For example, would it still be applicable to future models that potentially take more vision input and rely less on HTML and DOM? I acknowledge that it is a bit unfair to require this given the concurrent development of models, but it does limit the contribution of the work a bit.
> >
> > Overall, I agree that the work definitely has its merits, but I feel it is not enough for a conference as competitive as ICLR. I have increased the breakdown scores but kept my overall recommendation as marginal below acceptance.

---

> > > ### Author Response · Authors · 2024-11-26
> > > **Clarification on the reviewer response**
> > >
> > > We appreciate your feedback. We would like to clarify the following points:
> > >
> > > > "While it still shows an edge, it does not offer a conclusive answer on WebVoyager experiments"
> > >
> > > We would like to call out several points when considering this result. First, given that there are only 129 tasks, the overall success rate (SR; 54.3% for AgentOccam (ours) vs 51.9% for Agent-E baseline) is a much more reliable number with less variance than the per-website SR. Among the per-website SR, multiple websites have a very small number of total tasks (<10) and it is hard to compare two methods based on that. Second, the baseline approach, Agent-E, has a much more complex agent architecture (planner-browser-reflector) than ours, and is released on arxiv within 3 months of ICLR submission time which should be considered as concurrent work by ICLR policy. (According to [ICLR policy](https://iclr.cc/Conferences/2025/FAQ), a work is concurrent if they are published on a peer-reviewed venue within the last four months.) Thus, our understanding is that "better in 6, tied in 3, and worse in 4" is significant enough result compared to the SOTA method on WebVoyager (according to Agent-E paper).
> > >
> > > > "applicable to future models that potentially take more vision input"
> > >
> > > AgentOccam can incorporate visual information, but the structural description of web pages (HTML, DOM, etc.) is indispensable. Our code supports visual modal input using multimodal foundation models. However, the most common method now for vision—language—model—based web agents to integrate visual modality is the [set-of-mark](https://som-gpt4v.github.io), which involves segmenting visual elements and describing the information within the visual boxes in text. Considering that the integration of vision in current large models also heavily relies on textual descriptions, the effectiveness of integration greatly depends on the precision of these textual descriptions.
> > >
> > > Moreover, we would like to remark that the current multimodal foundation models' visual understanding and reasoning capabilities are relatively limited compared with their capabilities in text. Therefore, agents that rely purely on visual inputs (screenshots) perform much worse on the standard benchmarks. For example, the “[Tree Search for Language Model Agents](https://arxiv.org/abs/2407.01476)” methods use screenshots as observation and GPT-4o (the most competitive model on visual understanding) as the base model. However, its overall success rate on WebArena is 19.2%, which is significantly worse than ours and other methods primarily using text observations. **Thus, based on the current status of multimodal foundation models, we believe our approach is (one of) the most promising methods for building a performant agent.** If the reviewer’s questions are about generalizing to **future models** that do not yet exist, we acknowledge that this work is limited in that aspect since it is hard to predict the capabilities and pros and cons of future foundation models and the future of how websites will be developed, and design agent methods for that.
> > >
> > > > "Overall, I agree that the work definitely has its merits, but I feel it is not enough for a conference as competitive as ICLR."
> > >
> > > We sincerely thank that the reviewer agrees that our work **definitely has its merits**, but we are confused by why the reviewer thinks it is not **"enough"**. We believe our response above addressed the reviewer’s potential concerns in this new comment. In addition, here we would like to highlight why we think the key merit of our paper is enough, regardless of some of the concerns/questions we discussed.
> > >
> > > **We believe that our important technical contribution is a significantly simpler (without CoT, ICL examples, reflector, memory, more than one LLM call or module, etc) agent that outperforms most previous and concurrent agents on two non-trivial web benchmarks. This highlights a significant problem for LLM-based agents: the raw observation and action space in agentic tasks are not suitable for the current state-of-the-art LLMs’ capabilities.** This is also noticed in the release of many commercial models/agents, for example, "Some actions that people perform effortlessly—scrolling, dragging, zooming—currently present challenges for Claude..." from [Anthropic's Oct 22 release](https://www.anthropic.com/news/3-5-models-and-computer-use). We wish not to introduce many unnecessary complex algorithmic parts as a "novel contribution." In contrast, as our method's name highlighted, **we show that a powerful web agent can be as simple, low-cost, and effective as it is. We believe this argument is important to the current LLM agent community as more and more complex agent architectures come out every week or even every day.**

---

> > > > ### Comment · Reviewer_LC8w · 2024-11-30
> > > >
> > > > Thanks for the response. I agree that a proper action and observation space is definitely important, and as such many existing works have used some form of simplified space, e.g., a11y tree and limited actions. It is always hard to find a balance between simple actions that lead to higher success and complex actions that provide better coverage. I think the work has a contribution to specific web agent applications, but overall, I feel the technical novelty and potential impact for the broader community are still limited and a bit misaligned with ICLR.

---

### Official Review · Reviewer_NaFf · 2024-11-04

**Soundness:** 3
**Presentation:** 3
**Contribution:** 4
**Rating:** 8
**Confidence:** 4

**Summary:**

This work focuses on developing a simple generalized framework for LLM-based web agents, allowing them to leverage the strengths of LLMs to get complete web tasks. This framework focuses on aligning LLMs to the web domain and hence converts the problem to that of finding alignments/mappings in observation and action space between web domain and LLMs. As part of that, for action space alignment, they reduce actions to simple set of actions and add few additional "planning" related actions to the set. For observation space alignment, they propose to simplify the web page representation and also focus on selective representation of history for tracking. They show impressive results of the impact of these strategies on WebArena benchmark while also providing ablations to understand the impact of each of the recommended change. Finally, they show how this simple baseline agent can be promisingly combined with existing strategies.

**Strengths:**

The following are the major strengths of the paper:
-  They have focused on the first principles of building a domain-specific agent, i.e., a web agent and hence utilized domain-knowledge to optimized how LLMs can be used in the best way for this domain.
- The ablations presented are very well presented allowing the reader to understand the impact of each of the changes proposed in the paper.

**Weaknesses:**

- The results are shown on WebArena which is a simulated environment. It would be more realistic to see how it performs on WebVoyager which is more realistic in terms of how web behaves.
- The tree representations used for planning needs better explanation. I have to honest that I couldn't grasp it as well as I would want to. While I understand it in principle, presenting it with a more detailed example, even if in Appendix, might help someone like me look into the details of it.
- While the methodology/process presented in this paper of "optimizing/aligning LLMs to the domains when designing agents" is very important and valuable, it is worth noting that the specific proposed changes themselves (understandably) don't generalize to other domains beyond web.
- Minor typo in title of Sec 5 (not a weakness obviously but would be good to fix in the final version).

**Questions:**

- I would be curious to see how the agent performs on other benchmarks, especially WebVoyager. This is especially valuable benchmark as besides being more realistic, it will help answer my next question.
- The ideas presented in this work are extremely similar to another concurrent work on building a WebVoyager SOTA web agent, Agent-E. There is almost a one-to-one mapping between the proposed changes (both action and observational space) in AgentOccam and Agent-E.
i. The simplifying of action space in AgentOccam is close to the selection of "primitive actions" in Agent-E.
ii. The Planning tree in AgentOccam is close in principle to "hierarchical planning" in Agent-E.
iii. Simplifying Web Page observation in AgentOccam is same in principle to the idea behind "DOM distillation" in Agent-E.
iv. Selective replaying of observation history in AgentOccam is similar in principle to "change observation" in Agent-E.
And both agents are SOTA on different benchmarks, AgentOccam on WebArena and Agent-E on WebVoyager, and hence would be good to have a detailed comparison, both quantitative and qualitative, between them.

---

> ### Author Response · Authors · 2024-11-22
>
> We appreciate your insightful feedback and suggestions. We are glad to read that you find our alignment of observation and action space between web tasks and the training of large language models (LLMs) to be sound, pertinent to the current web agent literature, and contributing to the LLM-agent community. **We have updated the manuscript according to your feedback, and have added code for all experiments and new experiment trajectories to the supplementary materials.** We arrange our responses in the order your questions are raised, and hope they clarify your concerns.
>
> ## AgentOccam's Performance on Real-world Websites (Benchmark WebVoyager)
> As per your suggestion, we conduct experiments on the general-purpose web task benchmark WebVoyager and compare AgentOccam's performance with the benchmarks' state-of-the-art (SOTA) Agent-E. We kindly refer you to the general response section for the details. In short, AgentOccam performs better than Agent-E on the WebVoyager's definite-answer-subset. Specifically, both agents have their specificities. We find that Agent-E has strong performance on websites like Wolfram Alpha with more delicate plans issued by its "planner," and AgentOccam outperforms it on websites like Google Search and Hugging Face which need more accurate task interpretation and information retrieval. The impressive results across 10+ websites and many types of tasks can show AgentOccam's generalizability in the web domain.

---

> > ### Comment · Reviewer_NaFf · 2024-11-27
> >
> > Please see my response to this comparison here: https://openreview.net/forum?id=oWdzUpOlkX&noteId=5kkDiCqbkJ

---

> ### Author Response · Authors · 2024-11-22
>
> ## A Detailed Comparison of AgentOccam and Agent-E
> We appreciate that you bring Agent-E to our attention. We will include the discussion about this paper in the later versions, as well as the WebVoyager result table to show AgentOccam's web environment generalizability. We want to emphasize that while Agent-E might look similar to AgentOccam, there exists differences in those two agents' policy construction.
>
> First, Agent-E designs a planner-browser-reflector agent architecture on top of the system, and like many previous work, any compound agent architecture is susceptible to error propagation, which means the former module's erroneous behaviors would affect later performance (ii and iv). Though Agent-E cleverly finds a balance between the planner, browser, and reflector role by strategic prompting, many of its failures are induced by those agent roles' poor collaboration. For example, in Google Search task "Find the software requirements for iPhones that support AirDrop's ability to continue transmitting over the web when out of range." (Google Search--7), the planner starts by making a plan that requires the browser to navigate to Apple.com and find the answer there, complicating the problem and leading to failure. In contrast, AgentOccam simply types an appropriate query into the search field and finds the answer. Another example is the Cambridge Dictionary task "Search for the word 'sustainability' on the Cambridge Dictionary, what is the translation of sustainability into Chinese and French in the dictionary." (Cambridge Dictionary--6), where the reflector mistakes the Spanish translation of "sustainability" as its French translation, misleading the following agent roles' interaction and causing failure. Additionally, there are cases where the reflector accidentally terminates the trajectory when it wrongly thinks the task has ended. In summary, as different agent roles have varied responsibilities (e.g., reflector needs to provide verbal feedback) and control (e.g., planner can delegate navigation task to browser) over the trajectory flow, different roles' prompting biases are prone to induce specific misconduct modes, which will negatively affect the task completion sequentially and accumulatively. On the contrary, we keep AgentOccam's policy simplicity by using the same agent role (i.e., agent prompts) throughout the play, which, based on the empirical results, has the minimum misleading role-prompting biases compared to compound agent architectures.
>
> Second, AgentOccam and Agent-E are concurrent works and propose or implicitly follow many similar LLM-agent refinement principles inspired by those that have been proven useful in classical robotics learning (i and iii). Notably, AgentOccam introduces the concept of action alignment that enhances agent navigation focus by only retaining essential actions (i.e., click, type, note, stop, and go_back) in the agent action space, with each action space refinement component being shown helpful by ablation studies. Similarly, Agent-E pre-defines its submodule browser's skills to be click, enter text, open url, and press key (e.g., submit, pagedown). Moreover, AgentOccam proposes observation space alignment that revises the web environment observation to be more succinct (e.g., removing repetitive functionality- and layout-indicative tokens, reframing page content block in DOMs, organizing history with pivotal nodes and the planning tree) but as informative, turning out to both downsize the LLM consumed tokens and improve agent's performance by trajectory statistics and ablation studies. Likewise, Agent-E allows its browser to select from three types of DOMs (i.e., text-only, input fields, and all fields) that best suit the task. With Agent-E setting SOTA on WebVoyager and AgentOccam excelling on both WebArena and the WebVoyager subset, it follows that shared principles—specifically, action and observation space adaptations to the agent architecture design—enhance LLMs' understanding and reasoning on web tasks.

---

> > ### Comment · Reviewer_NaFf · 2024-11-27
> >
> > I would recommend against putting up the subset of results for the reasons raised here: https://openreview.net/forum?id=oWdzUpOlkX&noteId=5kkDiCqbkJ
> > Please refrain from putting up these results before manual evaluation across the entire dataset.

---

> ### Author Response · Authors · 2024-11-22
>
> ## Planning Tree Clarifications
> We will use a planning tree generated by our agent during the development stage as an example for explaining how the planning tree takes shape and helps in selectively replaying past pages. We have updated the paper in Section 4.2 and in the appendix with these clarifications to make it a clearer version.
>
> ### Intuition and Background
> If we make an analogy of the single-thread web task completion to code execution, the agent's issuing sub plans mimic nested function calling, the active plan is akin to the uppermost function in the execution stack, and pruning the planning tree plays involves popping the failed functions off the stack. There constantly exists a planning tree and the planning tree operation instructions in the agent's prompt. We enable the agent to organize the planning tree with `branch` and `prune` commands, where the `branch` action creates new subplans, and the `prune` action restores the task progress to a previous state. It's noteworthy that those planning commands are in the same position as navigation prompts for the agent.
>
> ### Prompt, Planning Tree Section
> ```bash
> If you think you should refine the plan, use the following actions:
> branch [parent_plan_id] [new_subplan_intent]: To create a new subplan based on PREVIOUS PLANS. Ensure the new subplan is connected to the appropriate parent plan by using its ID. E.g., `branch [12] [Navigate to the "Issue" page to check all the issues.]`
> prune [resume_plan_id] [reason]: To return to a previous plan state when the current plan is deemed impractical. Enter the ID of the plan state you want to resume. E.g., `prune [5] [The current page lacks items "black speaker," prompting a return to the initial page to restart the item search.]`
> Otherwise, use the following actions:
> {navigation_specifications}
> ```
>
> ### Example Task Objective
> (WebArena 174) Open my latest updated issue that has keyword "feature" in its title to check if it is closed.
>
> ### AgentOccam's Workflow Regarding the Planning Tree
> 0. At the beginning, we used the task objective as the root plan:
> ```
> [0] (Active Plan) Find the solution to "Open my latest updated issue that has keyword "feature" in its title to check if it is closed"
> ```
> 1. The agent added a subplan to planning node 0 by issuing `branch [0] [Navigate to the Issues page to search for the latest issue with the keyword "feature" in the title.]`. Now the planning tree changed into:
> ```
> [0] Find the solution to "Open my latest updated issue that has keyword "feature" in its title to check if it is closed"
> 	[1] (Active Plan) Navigate to the Issues page to search for the latest issue with the keyword "feature" in the title.
> ```
> 2. The agent navigated to the project's issue page.
> 3. The agent decomposed the plan by generating branch actions `branch [1] [Search for the latest issue with the keyword "feature" in the title and check if it is closed.]` and `branch [1] [Open the latest issue with the keyword "feature" in the title.]`. It performed navigation steps for each active plan before the next planning command was executed. In this example, after `branch [1] [Search for the latest issue with the keyword "feature" in the title and check if it is closed.]` was proposed, the active plan turned into "Search for the latest issue with the keyword "feature" in the title and check if it is closed." The agent then typed the keyword "feature" into the search box and sorted the issues by operating the sort icon before generating the next plan "Open the latest issue with the keyword "feature" in the title." In other words, all the navigation commands (e.g., search and sort) it issued were intended for the current active plan (e.g., search for the latest issue with the keyword "feature" in the title and check if it is closed). Just like a function call only needs to consider the function's scope, this allows the agent to only attend to the navigation actions dedicated to the active plan and the corresponding web observation as the playing history, which helps selectively replay past pages for the agent. Note that in this case, it assigned the two new sub plans to the same parent plan [1], which automatically shaped the planning tree's structure. Finally, the planning tree reformed into (the content enclosed in "[]" means comments, which is intended for illustration and didn't appear in the agent's prompt):
> ```
> [0] Find the solution to "Open my latest updated issue that has keyword "feature" in its title to check if it is closed"
> 	[1] Navigate to the Issues page to search for the latest issue with the keyword "feature" in the title. [Plan's action scope: navigate.]
> 		[2] Search for the latest issue with the keyword "feature" in the title and check if it is closed. [Plan's action scope: search and sort.]
> 		[3] (Active Plan) Open the latest issue with the keyword "feature" in the title.
> ```
> 4. The agent executed the following steps to complete the task.

---

> ### Author Response · Authors · 2024-11-22
>
> ## Miscs.
> Thank you for carefully checking our paper and pointing out the typo. We will fix it in the later versions. If you have any additional questions or concerns, please let us know so we can resolve them before the discussion period concludes. Otherwise, it would be greatly appreciated if you could raise your score to show that the existing concerns have been addressed. Thank you!

---

> ### Author Response · Authors · 2024-11-25
>
> Dear Reviewer,
>
> Thank you once again for your time and effort in reviewing our paper.
>
> As the discussion period is nearing its end, we kindly remind you that there are two days remaining for any further comments or questions. We would be grateful for the chance to address any additional concerns before it concludes.
>
> Thank you again!
>
> Best regards,
>
> Authors

---

> > ### Comment · Reviewer_NaFf · 2024-11-27
> >
> > Thank you for your detailed response. As such, I believe I have fairly assessed your work at the moment, and have given a fair score. Hence, I will keep it as-is. Thanks.

---

### Author Response · Authors · 2024-11-22
**Experiments on AgentOccam's Generalizability and Code and Trajectory Logs Update**

We appreciate all reviewers' feedback and recognition of AgentOccam's action and observation space alignment's contributions to the LLM-based web agent community. Here, we respond to the common questions raised by the reviewers. **We have updated the manuscript according to the feedback, and have added code for all experiments and new experiment trajectories to the supplementary materials.**

## AgentOccam is Generalizable to Real-world Websites
We additionally conduct experiments on the general-purpose web task benchmark WebVoyager and compare AgentOccam's performance with the benchmarks' SOTA Agent-E. The details are as follows:

WebVoyager benchmark (https://arxiv.org/abs/2401.13919) compiles web tasks from 15 popular real-world websites. It comprises tasks with two types of user questions: ones with "golden" answers that are definite and time-invariant, and ones with "possible" answers that are either open-ended with multiple potential answers or related to real-time information. We use WebVoyager questions with golden answers to avoid subjective human evaluations and exclude GitHub tasks, as the site's anti-scraping measures frequently cause page loading timeouts. Additionally, due to these measures, GitHub limits interactable elements, which prevents web page proper functionality, and the IP address hosting the agent is at risk of being banned. In contrast, WebArena's simulated GitLab environment mimics GitHub, enabling us to demonstrate AgentOccam's performance on similar tasks using existing results. In a nutshell, we have 129 tasks from WebVoyager with definite golden answers across 13 real-world web environments, including Allrecipes, Amazon, Apple, ArXiv, BBC News, Booking, Cambridge Dictionary, Coursera, ESPN, Google Map, Google Search, Huggingface, and Wolfram Alpha.

Our baseline on WebVoyager is the concurrent work Agent-E (https://arxiv.org/abs/2407.13032), which introduces several architectural improvements like the planner-browser-reflector agent architecture and the browser's document object model (DOM) selection. It achieves the previous SOTA on the full WebVoyager with the assessments done by humans. As they didn't report agent trajectory logs, we replicate their work on the WebVoyager subset introduced in the above paragraph, evaluated with the same definite-answer-based hard-coded evaluators as ours. We run each task one time for both agents. The detailed success rates (SRs) are as follows with the code and trajectory logs attached as the supplementary materials:

| Website (Task Number)               | Agent-E SR | AgentOccam SR |
|---------------------------------------|--------------------|-----------------------|
| **Allrecipes (4)**                    | 75.00%             | 50.00%               |
| **Amazon (1)**                        | 0.00%              | 0.00%                |
| **Apple (7)**                         | 57.14%             | 28.57%               |
| **ArXiv (16)**                        | 50.00%             | 43.75%               |
| **BBC News (2)**                      | 0.00%              | 0.00%                |
| **Booking (2)**                       | 50.00%             | 100.00%              |
| **Cambridge Dictionary (9)**          | 55.56%             | 88.89%               |
| **Coursera (2)**                      | 50.00%             | 50.00%               |
| **ESPN (10)**                         | 40.00%             | 50.00%               |
| **Google Map (9)**                    | 33.33%             | 44.44%               |
| **Google Search (16)**                | 62.50%             | 81.25%               |
| **Huggingface (17)**                  | 17.65%             | 29.41%               |
| **Wolfram Alpha (34)**                | 73.53%             | 61.76%               |
| **Overall (129)**                     | 51.9%              | 54.3%                |

Based on the results above, AgentOccam performs better than Agent-E on the WebVoyager's definite-answer-subset. Specifically, both agents have their specificities. We find that Agent-E makes an edge on websites like Wolfram Alpha with more delicate plans issued by its "planner," and AgentOccam outperforms it on websites like Google Search and Hugging Face with more accurate task interpretation and information retrieval. The impressive results across 10+ websites and many types of tasks can show AgentOccam's generalizability in the web domain.

---

> ### Author Response · Authors · 2024-11-22
> **Experiments on AgentOccam's Generalizability and Code and Trajectory Logs Update**
>
> ## AgentOccam is Generalizable to different LLM Families
> We conduct the full set of ablation studies on a WebArena development subset with Gemini-1.5-flash (https://blog.google/technology/ai/google-gemini-ai/#sundar-note), a model trained with different data from the GPT model family.
>
> Due to the time and cost constraints, we construct a representative subset from the original 812 tasks in WebArena. Specifically, we sample one task from each task cluster instantiated with the same intent template (e.g., “What is the top-{{n}} best-selling product in {{year}}”, where “{{n}}” and “{{year}}” would be replaced by instantiation tokens) in WebArena, forming a development set with 190 tasks. We use Gemini-1.5-flash for all experiments. We run each task one time and restart the experiment if the WebArena simulator fails (i.e., login expires, reddit post limit exceeds, map malfunctioning, etc.). We include the code and trajectory logs in the supplementary materials. The results are as follows, with the performance from the GPT-4-turbo counterpart on the same development task set for comparison:
>
> | Agent                       | Model         | SR (%) | Shopping | Shopping Admin | GitLab  | Map    | Reddit | Multisite |
> |-----------------------------|---------------|--------|----------|----------------|---------|--------|--------|-----------|
> | Vanilla                      | GPT-4-turbo       | 14.2   | 16.7     | 12.2           | 14.6    | 17.2   | 9.5    | 10.0      |
> | ↓ Actions                    | GPT-4-turbo     | 25.8   | 22.9     | 24.4           | 34.2    | 31.0   | 19.1   | 10.0      |
> | Above + X Scrolling          | GPT-4-turbo | 30.0   | 29.2     | 29.3           | 36.6    | 24.1   | 38.1   | 10.0      |
> | Above + Obs Opt.             | GPT-4-turbo | 34.7   | 37.5     | 34.2           | 22.0    | 41.4   | 57.1   | 10.0      |
> | Above + History              | GPT-4-turbo | 36.8   | 39.6     | 34.2           | 36.6    | 44.8   | 38.1   | 10.0      |
> | AgentOccam                   | GPT-4-turbo    | 44.2   | 45.8     | 46.3           | 48.8    | 41.4   | 47.6   | 10.0      |
> | Vanilla             | Gemini-1.5-flash        | 11.6   | 20.8     | 4.9            | 9.8     | 17.2   | 4.8    | 0.0       |
> | ↓ Actions           | Gemini-1.5-flash        | 23.2   | 29.2     | 22.0           | 29.3    | 13.8   | 19.1   | 10.0      |
> | Above + X Scrolling | Gemini-1.5-flash        | 24.2   | 33.3     | 24.4           | 22.0    | 27.6   | 14.3   | 0.0       |
> | Above + Obs Opt.    | Gemini-1.5-flash        | 30.0   | 37.5     | 34.2           | 31.7    | 20.7   | 23.8   | 10.0      |
> | Above + History     | Gemini-1.5-flash        | 32.1   | 35.4     | 31.7           | 31.7    | 37.9   | 28.6   | 10.0      |
> | AgentOccam          | Gemini-1.5-flash        | 33.7   | 37.5     | 34.2           | 36.6    | 37.9   | 28.6   | 0.0       |
>
> The main action statistics:
>
> | Exp.                       | Model                 | click | type | scroll | go_back | note | stop | go_home | branch | prune |
> |-----------------------------------|-----------------------|-------|------|--------|---------|------|------|---------|--------|-------|
> | Vanilla                           | GPT-4-turbo | 473   | 212  | 22     | 9       | -    | -  | -       | -      | -     |
> | ↓ Actions                         | GPT-4-turbo | 1612  | 515  | 107    | 16      | 40   | 120  | 5       | -      | -     |
> | Above + X Scrolling               | GPT-4-turbo | 1641  | 491  | -      | 19      | 60   | 128  | 5       | -      | -     |
> | Above + Obs Opt.                  | GPT-4-turbo | 1539  | 437  | -      | 21      | 82   | 137  | 1       | -      | -     |
> | Above + History                   | GPT-4-turbo | 1068  | 287  | -      | 24      | 23   | 181  | 24      | -      | -     |
> | AgentOccam                        | GPT-4-turbo | 1110  | 261  | -      | 123     | 40   | 183  | 3       | 9      | 6     |
> | Vanilla (Gemini)                  | Gemini-1.5-flash      | 103   | 65   | 7      | -       | -    | -  | -       | -      | -     |
> | ↓ Actions (Gemini)                | Gemini-1.5-flash      | 1390  | 509  | 176    | 28      | 31   | 130  | 11      | -      | -     |
> | Above + X Scrolling (Gemini)      | Gemini-1.5-flash      | 1258  | 542  | -      | 23      | 53   | 134  | 22      | -      | -     |
> | Above + Obs Opt. (Gemini)         | Gemini-1.5-flash      | 1322  | 377  | -      | 28      | 29   | 144  | 42      | -      | -     |
> | Above + History (Gemini)          | Gemini-1.5-flash      | 776   | 253  | -      | 50      | 44   | 185  | 15      | -      | -     |
> | AgentOccam (Gemini)               | Gemini-1.5-flash      | 942   | 289  | -      | 67      | 44   | 185  | 34      | 27     | 75    |

---

> ### Author Response · Authors · 2024-11-22
> **Experiments on AgentOccam's Generalizability and Code and Trajectory Logs Update**
>
> ## AgentOccam is Generalizable to different LLM Families (Contd.)
> The average observation tokens per step:
>
> | Exp.                       | Model                 | ALL   | SHOPPING | SHOPPING_ADMIN | GITLAB | MAP   | REDDIT | MAP   | MULTISITE |
> |-----------------------------------|-----------------------|-------|----------|----------------|--------|-------|--------|-------|-----------|
> | Vanilla                           | GPT-4-turbo | 2202.5| 2268.7   | 2421.6         | 2267.3 | 1882.3| 2119.0 | 1882.3| 1715.8    |
> | ↓ Actions                         | GPT-4-turbo | 1682.1| 1496.7   | 2186.6         | 2168.3 | 953.3 | 1102.3 | 953.3 | 1261.8    |
> | Above + X Scrolling               | GPT-4-turbo | 3571.2| 3120.3   | 5175.5         | 4234.9 | 1423.3| 3079.8 | 1423.3| 1977.9    |
> | Above + Obs Opt.                  | GPT-4-turbo | 2669.8| 1664.9   | 4274.6         | 2744.3 | 1930.8| 2664.4 | 1930.8| 1370.6    |
> | Above + History                   | GPT-4-turbo | 3107.0| 1745.4   | 4988.5         | 3579.2 | 788.0 | 2758.4 | 788.0 | 1856.9    |
> | AgentOccam                        | GPT-4-turbo | 2785.1| 1500.8   | 5032.8         | 3032.3 | 645.6 | 3476.0 | 645.6 | 1233.9    |
> | Vanilla                  | Gemini-1.5-flash      | 2303.4| 2307.0   | 2537.5         | 2707.0 | 1849.7| 1794.7 | 1849.7| 2037.6    |
> | ↓ Actions                | Gemini-1.5-flash      | 1713.9| 1577.4   | 2112.3         | 2257.1 | 818.5 | 1542.8 | 818.5 | 1177.2    |
> | Above + X Scrolling      | Gemini-1.5-flash      | 3219.6| 3110.7   | 5234.0         | 3378.0 | 1040.7| 3204.9 | 1040.7| 2021.1    |
> | Above + Obs Opt.         | Gemini-1.5-flash      | 2814.9| 1769.8   | 4632.9         | 3410.7 | 858.7 | 3203.5 | 858.7 | 1577.0    |
> | Above + History          | Gemini-1.5-flash      | 3283.2| 1720.7   | 5078.3         | 3657.6 | 734.2 | 5712.5 | 734.2 | 1112.4    |
> | AgentOccam               | Gemini-1.5-flash      | 2872.2| 1639.6   | 4156.3         | 2519.3 | 688.7 | 6241.4 | 688.7 | 1267.7    |
>
> The average number of steps per task:
>
> | Exp.                       | Model                 | ALL  | SHOPPING | SHOPPING_ADMIN | GITLAB | MAP  | REDDIT | MULTISITE |
> |-----------------------------------|-----------------------|------|----------|----------------|--------|------|--------|-----------|
> | Vanilla                           | GPT-4-turbo | 1.3  | 1.5      | 1.3            | 1.1    | 1.4  | 1.3    | 0.9       |
> | ↓ Actions                         | GPT-4-turbo | 3.0  | 2.9      | 3.1            | 3.2    | 3.3  | 2.4    | 2.4       |
> | Above + X Scrolling               | GPT-4-turbo | 2.9  | 2.2      | 3.3            | 3.2    | 3.5  | 2.2    | 2.8       |
> | Above + Obs Opt.                  | GPT-4-turbo | 2.7  | 2.4      | 2.7            | 3.4    | 2.7  | 2.4    | 2.7       |
> | Above + History                   | GPT-4-turbo | 2.0  | 1.3      | 2.4            | 2.4    | 1.8  | 1.4    | 3.2       |
> | AgentOccam                        | GPT-4-turbo | 2.1  | 1.7      | 2.2            | 2.4    | 2.2  | 2.0    | 3.0       |
> | Vanilla                  | Gemini-1.5-flash      | 0.5  | 0.5      | 0.4            | 0.5    | 0.5  | 0.3    | 0.6       |
> | ↓ Actions                | Gemini-1.5-flash      | 2.8  | 3.2      | 2.8            | 2.5    | 2.6  | 2.9    | 2.4       |
> | Above + X Scrolling      | Gemini-1.5-flash      | 2.5  | 2.4      | 2.2            | 2.4    | 2.7  | 2.7    | 3.3       |
> | Above + Obs Opt.         | Gemini-1.5-flash      | 2.4  | 2.5      | 2.3            | 2.8    | 2.4  | 1.9    | 2.2       |
> | Above + History          | Gemini-1.5-flash      | 1.6  | 1.6      | 1.7            | 1.7    | 1.4  | 1.5    | 1.7       |
> | AgentOccam               | Gemini-1.5-flash      | 2.0  | 1.6      | 2.4            | 2.0    | 2.2  | 1.8    | 2.8       |
>
> We can observe a similar trend with the Gemini model that each alignment component introduced by our paper contributes to the web agent’s overall performance. To be specific, removing non-essential actions (↓ Actions) encourages the agent to explore more actively with commands `click` and `type`; disabling scrolling (Above + X Scrolling) proves advantageous in tasks where key information is not presented on the first browser sight; simplifying web page elements (Above + Obs Opt.) reduces the observation token number; selectively replaying web element in one page (Above + History) reduces the steps required to accomplish the task; and planning and selectively replay past pages (AgentOccam) enables the agent to self-organize task workflow and quickly restore to a previous stage after several failed attempts. In conclusion, each alignment component proposed by AgentOccam proves beneficial to the agent system and can generalize to different LLMs.

---

> ### Comment · Reviewer_NaFf · 2024-11-27
>
> As a general comment, your evaluation between Agent-E and your results are not a fair comparison for two reasons:
> - You are evaluating only on a subset as you mentioned.
> - The subset that you are measuring is the ones with "golden" subset that seemingly has deterministic answer. However, that assumption is wrong, for example, one of the ESPN task is (ID: ESPN--17) :
> "Check out the NBA Basketball Power Index 2023-24 to see which teams are in first place and which are in last place."
> The "golden answer" is "Boston Celtics; San Antonio Spurs". However, this is not true (https://www.espn.com/nba/bpi/_/season/2024) and there are 10s of such "wrong" golden answers. When evaluating in the above comparison table, for example, Agent-E performs poorly on ESPN in comparison to AgentOccam, but it would have been considered wrong according to this golden answer. Hence, a fair comparison would be with a full set and using manual annotation. These results should not be published until such a rigorous evaluation is performed.

---

> ### Author Response · Authors · 2024-11-30
>
> Thank you for the additional comments. We respectfully disagree that the comparison is unfair; please find our arguments below. Additionally, on the reviewer’s insistence, we performed a thorough manual evaluation, the process and results for the same are included below as well.
>
> > On Selection of 129 Tasks
>
> For benchmarks with a large volume of tasks, it is a common practice to test a statistically significant subset of tasks for assessment efficiency. For instance, [VisualAgentBench](https://arxiv.org/abs/2408.06327) introduced WebArena-Lite, and [WorkArena](https://arxiv.org/abs/2403.07718) also selected a limited set of tasks for testing on their proposed benchmark. Our selection of 129 subtasks is entirely based on the golden-answer labels from the WebVoyager benchmark dataset, as detailed in their [dataset repository](https://github.com/MinorJerry/WebVoyager/blob/main/data/reference_answer.json). **This categorization and labels are not introduced by us, thus eliminating any subjective task selection bias or unfairness to either our or the baseline method. We chose to evaluate on this subset because of a shortage of time during the rebuttal phase. We would be happy to include results on all the 643 tasks for the camera ready.**
>
> > On Incorrect Evaluators in WebVoyager
>
> Firstly, we would like to point out that any issues with the benchmark's labels would likely affect all agents' performance evaluations similarly, making the success rate figures a statistically fair reflection of agent performance. **However, in the spirit of a thorough evaluation, we performed manual labeling of trajectories from the 129 tasks with the help of three different human annotators. We observe that all human assessments show improvements in the performance of both agents compared to hard-coded evaluators (with improvements not exceeding 5%), with AgentOccam consistently leading by a slight margin.** The labeling details are described below.
>
> **In summary, we argue that our evaluation supports a statistically significant "fair comparison."** We want to clarify that we did not directly compare with the results from the Agent-E paper. Instead, we replicated the method on the same subset, using the same reference answers and evaluators/annotators as assessing our method. **As noted above, neither subset selection nor errors from the original benchmark are introduced or leveraged by either agent policy. Therefore, they should be orthogonal to the agent performance comparison and pose no bias towards either method.**

---

> ### Author Response · Authors · 2024-11-30
>
> ## Labeling Details:
> We asked three human labelers to mark the correctness of trajectories, two of whom (Alice and Bob) had no prior involvement in web agent projects. We provided them with the WebVoyager questions, the start URLs, and the responses to the WebVoyager questions by Agent-E and AgentOccam (**with the agents' identities concealed**), asking them to give a binary success/failure evaluation and their answers if they were to respond. They were also provided the reference answer link from WebVoyager and warned that the answers might contain errors. To avoid biasing their choice with the agent's identity, for questions difficult to judge based solely on the responses, we only provided details of the agent trajectory logs at the end, as needed by the human labelers. We would like to, but we cannot add more to the supplementary materials at this moment due to the rebuttal schedule. We will attach the assessment files to the supplementary materials once policy allows. The evaluation statistics are detailed as follows:
>
>
> | Website | Agent-E (Alice) | AgentOccam (Alice) | Agent-E (Bob) | AgentOccam (Bob) | Agent-E (Colin) | AgentOccam (Colin) | Avr Agent-E (%) | Avr AgentOccam (%) |
> |---------------------------|--------------------------|--------------------------|--------------------------|--------------------------|--------------------------|--------------------------|-----------|-----------|
> | Allrecipes | 75.00% | 50.00% | 75.00% | 75.00% | 75.00% | 75.00% | 75.00% | 66.67% |
> | Amazon | 0.00% | 0.00% | 0.00% | 0.00% | 0.00% | 0.00% | 0.00% | 0.00% |
> | Apple | 57.14% | 42.86% | 71.43% | 42.86% | 57.14% | 42.86% | 61.57% | 42.86% |
> | ArXiv | 56.25% | 62.50% | 56.25% | 50.00% | 62.50% | 50.00% | 58.33% | 54.17% |
> | BBC News | 50.00% | 50.00% | 0.00% | 0.00% | 0.00% | 0.00% | 16.67% | 16.67% |
> | Booking | 50.00% | 50.00% | 50.00% | 50.00% | 50.00% | 50.00% | 50.00% | 50.00% |
> | Cambridge Dictionary | 66.67% | 77.78% | 55.56% | 88.89% | 55.56% | 77.78% | 59.57% | 81.48% |
> | Coursera | 50.00% | 50.00% | 50.00% | 50.00% | 50.00% | 50.00% | 50.00% | 50.00% |
> | ESPN | 50.00% | 40.00% | 40.00% | 30.00% | 40.00% | 30.00% | 43.33% | 33.33% |
> | Google Map | 44.44% | 44.44% | 22.22% | 44.44% | 33.33% | 55.56% | 33.33% | 48.15% |
> | Google Search | 68.75% | 87.50% | 62.50% | 81.25% | 75.00% | 81.25% | 68.75% | 82.14% |
> | Huggingface | 29.41% | 41.18% | 23.53% | 47.06% | 23.53% | 41.18% | 25.82% | 43.47% |
> | Wolfram Alpha | 67.65% | 64.71% | 79.41% | 67.65% | 73.53% | 67.65% | 73.53% | 66.47% |
> | Total | 56.59% | 58.91% | 55.04% | 58.14% | 55.81% | 57.36% | 55.81% | 58.14% |
>
>
> The overall impressions of the two web agents by the three reviewers are copied below (for ease of reading, we have replaced references to Agent A and Agent B with their actual identities):
>
> Alice: They make quite endearing mistakes. Overall, AgentOccam seems better because, although it often claims it cannot find an answer and makes some humorous errors, Agent-E returns incorrect answers.
>
>
> Bob: The most direct feeling is that AgentOccam has less redundant information. Agent-E always seems to give everything it knows to avoid errors, like in a question that only asked for a number, Agent-E, as if not understanding, still listed a lot.
>
>
> Colin: Agent-E's answers tend to be longer, while AgentOccam's are more concise. Agent-E's answers require careful examination as they might contain errors, whereas AgentOccam simply states it does not know.
>
> Note: AgentOccam's statement that it cannot find an answer is due to us applying the same configuration from WebArena, which limits the step count to 20 steps; this limitation does not affect Agent-E's performance. Observing some of AgentOccam's failed trajectories reveals that tasks are interrupted midway toward success due to the step limit. If required for the camera-ready version to complete the WebVoyager benchmark, we will rerun these 129 experiments, relaxing AgentOccam's (ours) step limit.

---

### Author Response · Authors · 2024-12-02

We express our gratitude to all reviewers for their feedback, which has enriched our work. We are delighted to see that the reviewers have identified several strengths in our paper, including:

* The paper proposes a simple yet effective LLM-based web agent (NaFf, LC8w, Tca3);
* The idea of aligning the observation and action representation of web tasks with LLM capabilities is crucial and inspiring (NaFf, Tca3), offering a new pathway that could extend to other complex tasks (Tca3);
* The results show impressive improvements over the plain base agent and the state-of-the-art on the WebArena benchmark (NaFf, LC8w, Tca3);
* The ablation studies are comprehensive (NaFf, LC8w);
* Our simple baseline agent can be promisingly combined with existing strategies, e.g., LLM-as-a-judge (NaFf, LC8w).

Upon reviewers' suggestions, we have included additional experiments that show our web agent is generalizable to real-world websites and different LLM families. We have also added code for all experiments and new experiment trajectories to the supplementary materials.

We are grateful for the opportunity to discuss our work with the reviewers and are enthusiastic about the potential impacts of our contributions to the field. Thank you for your encouragement and for recognizing our efforts.

---

### Meta-Review · Area_Chair_xf5Y · 2024-12-16

**Metareview:**

This work introduces AgentOccam, an approach designed to align the action and observation spaces of large language model (LLM) agents for web-based tasks. Specifically, the method features an automated pipeline for pruning web page content and defining a more concise action space, optimizing the agent’s interaction efficiency. The proposed approach achieves impressive performance on the WebArena benchmark and demonstrates generalization capabilities through evaluations in live web environments using WebVoyager.

A notable limitation of this work is its focus on web agents that operate solely on text-based representations of web pages, such as HTML or accessibility trees. While effective in its scope, this constraint may limit its applicability to more visually complex or non-text-dominant web interfaces.

**Additional Comments On Reviewer Discussion:**

The authors added more experiments on more benchmarks during rebuttal, but the reviewers maintained the original score.

---

### Decision · Program_Chairs · 2025-01-22

Accept (Poster)